# Beyond Interpretability: The Gains of Feature Monosemanticity on Model Robustness

**Qi Zhang**[1][*] **Yifei Wang**[2][*] **Jingyi Cui**[1] **Xiang Pan**[3]
**Qi Lei**[3] **Stefanie Jegelka**[4,5] **Yisen Wang**[1,6][†]

[1] State Key Lab of General Artificial Intelligence,
 School of Intelligence Science and Technology, Peking University
[2] MIT CSAIL
[3] New York University
[4] TUM CIT, MCML, MDSI
[5] MIT EECS, CSAIL
[6] Institute for Artificial Intelligence, Peking University

## Abstract

Deep learning models often suffer from a lack of interpretability due to *polysemanticity*, where individual neurons are activated by multiple unrelated semantics, resulting in unclear attributions of model behavior. Recent advances in *monosemanticity*, where neurons correspond to consistent and distinct semantics, have significantly improved interpretability but are commonly believed to compromise accuracy. In this work, we challenge the prevailing belief of the accuracy-interpretability tradeoff, showing that monosemantic features not only enhance interpretability but also bring concrete gains in model performance of robustness-related tasks. Across multiple robust learning scenarios—including input and label noise, few-shot learning, and out-of-domain generalization—our results show that models leveraging monosemantic features significantly outperform those relying on polysemantic features. Furthermore, we provide empirical and theoretical understandings on the robustness gains of feature monosemanticity. Our preliminary analysis suggests that monosemanticity, by promoting better separation of feature representations, leads to more robust decision boundaries under noise. This diverse evidence highlights the **generality** of monosemanticity in improving model robustness. As a first step in this new direction, we embark on exploring the learning benefits of monosemanticity beyond interpretability, supporting the long-standing hypothesis of linking interpretability and robustness. Code is available at https://github.com/PKU-ML/Monosemanticity-Robustness.

## 1 Introduction

A long-standing problem of deep learning is the so-called "black-box" nature. People find that an important factor for its lack of interpretability is feature *polysemanticity*, where a single neuron (a dimension of feature maps) is activated by multiple *irrelevant* semantics (Arora et al., 2018; Olah et al., 2020), preventing clear attributions of neural behaviors. Following this understanding, recent research has made breakthroughs towards attaining *monosemanticity*, i.e., neurons corresponding to consistent semantics (monosemantic), which dramatically improves model interpretability; see an illustrative example on polysemanticity and monosemanticity in Figure 1(a). They achieve this through architectural designs (Elhage et al., 2022a; Wang et al., 2024) or post-training explanation modules (Cunningham et al., 2024), and have successfully scaled to visual backbones (*e.g.,* ResNet) and large language models (LLMs) (*e.g.,* Claude, GPT, and Gemma), discovering many intriguing phenomena and applications (Templeton, 2024; Gao et al., 2024; Lieberum et al., 2024).

However, these works on monosemanticity suggest an inevitable "accuracy-interpretability" tradeoff: monosemantic features, although more interpretable, come at the sacrifice of expressive power and

---

[*]Equal Contribution.
[†]Corresponding Author: Yisen Wang (yisen.wang@pku.edu.cn).

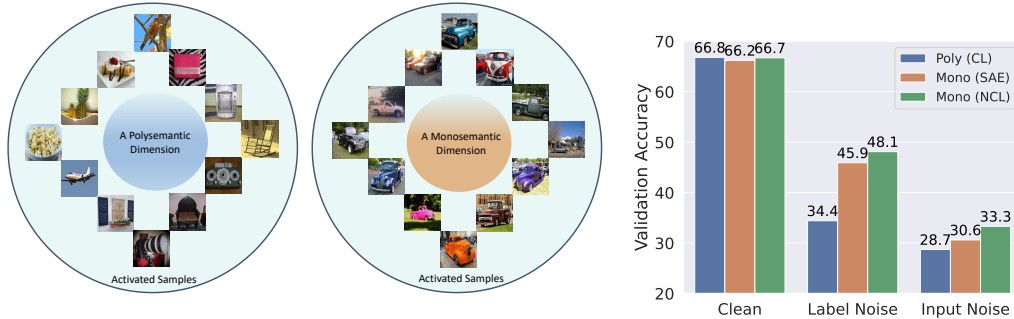

(a) Illustration of activated samples on a Polysemantic (Left) and a Monomsemantic (Right) dimension

(b) Test accuracy (%) of classifiers learned upon Polysemantic and Monosemantic features on different scenarios

Figure 1: A comparison between Polysemantic (Contrastive Learning (CL)) and Monosemantic features (Non-negative Contrastive Learning (NCL), Sparse Autoencoder (SAE)) pretrained on ImageNet-100. We consider noisy labels (90% noise rate) and Gaussian input noise (0.6 stdev); see more details in Appendix A.4.

underperform polysemantic features at prediction accuracy. For example, Gao et al. (2024) observe that the sparse autoencoders significantly enhance the monosemanticity of language models while obtaining inferior performance in downstream prediction tasks. Elhage et al. (2022b) show that monosemantic representations indicate lower model performance in reconstruction tasks with a toy model. Cunningham et al. (2024) exhibit a negative correlation between the expressive power and monosemanticity. This widely accepted belief limits the applications of monosemanticity techniques to only interepretability-related domains. In this paper, we aim to push this boundary one step forward by demonstrating that monosemanticity can also bring significant gains on practical model performance beyond interpretability.

In particular, we discover a widely appearing phenomenon, that **monosemantic features are much more robust** compared to polysemantic features, across multiple scenarios related to "robustness". One such scenario is **learning with noise**. Real-world data are often imperfect with low-quality input and mislabeling, manifested in the form of various data noises and distribution shifts. We find that under either input or label noises, learning a classifier upon (pretrained) monosemantic features can attain much higher accuracy (*e.g.,* +13.7% top-1 accuracy under 90% label noise) than polysemantic features, as shown in Figure 1(b). This feature-centric result also offers a new perspective to noisy learning where existing studies primarily focus on robust learning objectives (Wang et al., 2019a; Song et al., 2020; Ma et al., 2020).

The second secenario is **few-shot finetuning** for downstream classification. Today's large visual backbones often need to be finetuned on a small amount of downstream labeled data, where models easily overfit and deteriorate. We find that *monosemantic finetuning, i.e.,* preserving the monosemanticity of representations during finetuning (with a technique from Wang et al. (2024)), can attain much higher accuracy under few-shot data compared to vanilla *polysemantic finetuning* (*e.g.,* +3.9% top-1 accuracy with 10% samples). The same method also works for finetuning with noisy data or training from scratch.

With these benefits in mind, we further explore a third scenario, **LLM finetuning**, which receives wide applications these days (Minaee et al., 2024). Pretrained LLMs need to be carefully finetuned on small-scale language data for different purposes, *e.g.,* instruction following and certain abilities (*e.g.,* reasoning), while avoiding conflicting and forgetting. Since LLMs do not have a natural representation space like visual models, we devise a simple sparse variant of LoRA (an efficient tuning method), named **MonoLoRA**, to encourage the monosemanticity of the updates of all features. We show preliminary evidence that when finetuning an aligned LLM (Llama-2-7b-chat) on SST-2 (a classification task) and Dolly (instruction following task), MonoLoRA better preserves model alignment while improving task performance.

At last, we provide a deeper understanding of the robustness gains of monosemanticity. Empirically, we compare the salient features of different classifiers, observing that the more robust classifiers tend to depend on more monosemantic features. Theoretically, as a preliminary step, we compare polysemantic and monosemantic features under a toy model proposed in Elhage et al. (2022b). Our

analysis suggests that monosemantic features, due to their better separation, are less susceptible to overfitting to noise, resulting in more robust decision boundaries than polysemantic features.

In summary, this work challenges the common "accuracy-interpretability" tradeoff by demonstrating the potential of feature monosemanticity to bring clear gains in model accuracy of robustness-related tasks. These gains manifest themselves in various aspects of "learning robustness" that we can think of: input noise, label noise, out-of-domain data, few-shot image data, and few-shot language data. The diverse set of evidence strongly indicates that feature monosemanticity provides *a general sense of robustness* compared to polysemantic features, echoing with the long-lasting hypothesis on the relationship between better feature interpretability and better robustness (*e.g.,* human decisions are both interpretable and robust) (Bengio et al., 2013; 2019). As a first step in this direction, we believe that it will embark on more intriguing discoveries and understandings on the learning benefits of monosemanticity beyond interpretability.

## 2 PRELIMINARY & RELATED WORK

**Polysemanticity and Superposition Hypothesis.** Across various domains, many previous studies (Nguyen et al., 2016; Mu & Andreas, 2020; Olah et al., 2020) have consistently observed that a feature dimension in neural networks is usually activated with multiple unrelated semantics. Researchers define this phenomenon as the feature polysemanticity. In contrast, when each dimension is activated with a single latent natural concept, the features are denoted as monosemantic features. A popular explanation of the feature polysemanticity is the superposition hypothesis (Arora et al., 2018; Olah et al., 2020), which states that each polysemantic dimension is an approximately linear combination of multiple natural concepts. To verify that, Elhage et al. (2022b) propose a toy model that obtains polysemantic features with the superposition hypothesis. Comparing polysemantic and monosemantic features, there exists a common belief that monosemantic features exhibit better interpretability at the cost of downstream performance (Cunningham et al., 2024; Elhage et al., 2022b). However, in this paper, we challenge this trade-off, finding that monosemantic features also show superiority when the performance is evaluated on robustness tasks.

**Methods to Attain Feature Monosemanticity.** To enhance the feature interpretability, researchers propose several methods to obtain monosemantic features. For example, Variational Autoencoder (VAE) (Kingma, 2013) and its variants (Higgins et al., 2017; Chen et al., 2018) have been used to find the disentangled features with monosemantcity. However, the performance of these methods in real-world tasks like image classification and natural language understanding is quite unsatisfactory. Recently, researchers have tried to attain monosemanticity with minimal influence on performance. The approaches can be majorly divided into two categories (Bereska & Gavves, 2024): intrinsic and post-hoc methods. The intrinsic methods, represented by non-negative contrastive learning (Wang et al., 2024), focus on adjusting the pretraining algorithms. While the post-hoc methods apply downstream modifications on learned features. For example, the sparse autoencoder, which reconstructs the features from a sparse bottleneck layer, has recently shown impressive monosemanticity in various models (Ng et al., 2011; Gao et al., 2024). Different from existing works, our paper 1) analyzes the monosemanticity from a totally new perspective, i.e., we observe the relationship between monosemanticity and robustness; 2) challenges the widely-believed interpretability-accuracy trade-off in realistic and widely appearing scenarios; 3) introduces a new theoretical perspective to analyze the monosemanticity; 4) proposes new implementations to enhance monomsenaticity in large language models.

**Robustness Learning.** In the development of deep learning models, robustness is a critical measure for evaluating the quality of features (Wang et al., 2021; Xu et al., 2021). The evaluation of robustness involves various task scenarios. Common conditions include assessing the robustness of features against noisy labels (Song et al., 2022; Ma et al., 2018; Wang et al., 2018), distribution shifts (Yang et al., 2024; Wang et al., 2022; Xin et al., 2023), overfitting (Ying, 2019; Wang et al., 2019b), etc. In this paper, we analyze the robustness from a new perspective, i.e., we evaluate the influence of monosemanticity in different robustness tasks. For learning with noisy labels, we apply the symmetric label noise to the training samples, i.e., with a probability $\eta$ (noise rate), the labels of samples are uniformly flipped to the other classes. For robustness against distribution shifts, we apply various shifts, such as Gaussian noise, uniform noise, and real-world distribution shifts (Wang et al., 2019a; Geirhos et al., 2018) to the validation samples. For robustness against overfitting, we finetune the vision and language models with fewer samples and evaluate the validation performance.

## 3 THE ROBUSTNESS GAINS OF MONOSEMANTICITY

In this section, we compare polysemantic with monosemantic features across three different robust learning scenarios commonly encoutered **in the foundation model regime**: **first**, noisy linear probing on pretrained features (either polysemantic or monosemantic); **second**, noisy and few-shot finetuning from pretrained weights; **third**, finetuning LLMs on small-scale supervised data.

### 3.1 MONOSEMANTIC FEATURES ARE ROBUST UNDER LINEAR PROBING

Foundation models typically have two training phases: 1) *self-supervised learning* (SSL) on massive unlabeled data, and 2) *supervised finetuning* on small human-labeled data (classification, instruction following, or specific tasks). In fact, since SSL-pretrained features contain rich semantics, learning a simpler linear classifier on top, known as **linear probing** (LP), can often attain competitive performance to fully supervised ones (Chen et al., 2020). Therefore, we start with this simplest setting for comparing the robustness of polysemantic and monosemantic *pretrained* features. Specifically, we consider a standard linear probing setting, where we first pretrain features on unlabeled data and then learn a linear classifier on top with noisy labeled data.

#### 3.1.1 METHODS FOR FEATURE MONOSEMANTICITY

Among existing interpretability research, there are two categories of methods to attain monosemanticity: 1) intrinsic methods, where pretrained features are intrinsically monosemantic; 2) post-hoc methods, where we apply additional techniques to decode (polysemantic) pretrained features to monomsemantic ones. Here, we consider two representative methods for each paradigm.

**Intrinsic Monosemanticity with NCL.** Many previous works have tried to train interpretable features by adding sparsity regularization (Tibshirani, 1996) or identifiability constraints (Zhang et al., 2024); but they hardly scale to large-scale data with competitve performance. A recent work, NCL (non-negative contrastive learning) (Wang et al., 2024), as a modern counterpart to NMF (non-negative matrix factorization) (Lee & Seung, 1999), attains high sparsity and monosemanticity while having minimal influence on final performance. Specifically, NCL adopts the following InfoNCE loss (Oord et al., 2018) with non-negative feature outputs:

$$\mathcal{L}_{\mathrm{NCL}}(f) = -\mathbb{E}_{x,x^+} \log \frac{\exp(f_+(x)^\top f_+(x^+))}{\exp(f_+(x)^\top f_+(x_+)) + \frac{1}{M}\sum_{i=1}^{M}\exp(f_+(x)^\top f_+(x_i^-))}, \quad (1)$$

where $(x, x^+)$, $(x, x^-)$ are the positive and negative pairs in contrastive learning, $f_+(x) = \sigma(f(x))$, $\sigma$ is an activation function and $f$ is the original neural network. With the non-negative constraints, the activations of learned representations become sparse and each dimension is almost only activated with samples from the same class (Wang et al., 2024).

**Post-hoc Monosemanticity with SAE.** Another approach is to apply downstream modification on pretrained neural networks. Sparse autoencoders (SAEs) (Ng et al., 2011) find wide success in attaining monosemanticity in language models (Templeton, 2024; Gao et al., 2024; Lieberum et al., 2024). SAEs reconstruct the original outputs of pretrained networks from a sparse bottleneck layer. To be specific, the encoder and decoder are defined as:

$$\begin{aligned} z(x) &= \mathrm{topK}((W_{\mathrm{enc}}(f(x) - b_{\mathrm{pre}})), \\ \hat{f}(x) &= W_{\mathrm{dec}} z(x) + b_{\mathrm{pre}}, \end{aligned} \quad (2)$$

where $f(x)$ is the representation of input $x$; $W_{\mathrm{enc}}$, $W_{\mathrm{dec}}$, $b_{\mathrm{pre}}$ and $b_{\mathrm{enc}}$ are the parameters of SAE; $\mathrm{topK}$ is a sparse activation function proposed by Gao et al. (2024) that only preserves the top $K$ elements; and the SAE training loss is the reconstruction of original features, i.e., $\mathcal{L}_{\mathrm{SAE}} = \mathbb{E}_x \|\hat{f}(x) - f(x)\|^2$. As a result, the sparse latent feature $z(x)$ has much better monosemanticity than the original feature $f(x)$.

**Evaluation of Monosemanticity.** We evaluate the monosemanticity of different models with current metrics. Specifically, we follow Wang et al. (2024) and adopt semantic consistency as the metric. The semantic consistency calculates the proportion of activated samples that belong to their most frequent class along each dimension. We evaluate the monosemanticity of different models, including original models, models trained by NCL, and models with SAEs. As shown in Table 1, both NCL and SAE significantly improve the semantic consistency of trained models.

Table 1: Semantic consistency of different models. The semantic consistency calculates the proportion of activated samples that belong to their most frequent class along each dimension.

| Models | Poly (CL) | Mono (NCL) | Mono (SAE) |
|---|---|---|---|
| Linear Probing on CIFAR-100 | 1.0 | **8.2** | 3.1 |
| Linear Probing on Imagenet-100 | 1.0 | **12.3** | 7.2 |
| | Poly (CE) | Mono (NCE) | |
| Fine-tuning with 20% Samples | 2.1 | **18.4** | |
| Fine-tuning with 20% Noisy Lables | 3.4 | **20.6** | |

Table 2: Linear probing accuracy and gain (%) of polysemantic and monosemantic representations on ImageNet-100 and CIFAR-100 under different rates (%) of label noise (0 (clean label) to 90).

| Dataset | Features | 0 | 10 | 30 | 60 | 90 |
|---|---|---|---|---|---|---|
| | Poly | **54.6±0.2** | 53.2±0.2 | 52.1±0.4 | 49.5±0.2 | 35.4±0.3 |
| | Mono(SAE) | 54.5±0.1 | 53.9±0.2 | **53.0±0.3** | 50.6±0.4 | 38.0±0.1 |
| CIFAR-100 | | -0.1 | +0.7 | +0.9 | +1.1 | +2.6 |
| | Mono(NCL) | 52.8±0.4 | **54.2±0.3** | 52.7±0.2 | **51.5±0.2** | **45.0±0.2** |
| | | -1.8 | +1.0 | +0.6 | +2.0 | +9.6 |
| | Poly | **66.8±0.2** | 63.3±0.2 | 60.1±0.2 | 54.9±0.4 | 34.5±0.2 |
| | Mono(SAE) | 66.1±0.2 | 65.2±0.2 | 60.7±0.2 | 58.6±0.1 | 45.7±0.2 |
| ImageNet-100 | | -0.7 | +1.9 | +0.6 | +3.7 | +11.2 |
| | Mono(NCL) | 66.8±0.2 | **65.5±0.1** | **63.8±0.2** | **60.8±0.2** | **48.1±0.2** |
| | | +0.0 | +2.2 | +3.7 | +5.9 | +13.6 |

### 3.1.2 EXPERIMENTS

**Setup.** For the baseline, we pretrain a ResNet-18 (He et al., 2016) backbone with the widely-used contrastive framework SimCLR (Chen et al., 2020) on CIFAR-100 and ImageNet-100. In comparison, we use Non-negative Contrastive Learning (Wang et al., 2024) and Sparse Autoencoder (Gao et al., 2024) to represent two primary strategies for obtaining monosemantic features, i.e., improve the pretraining algorithm and apply downstream modification. For Non-negative Contrastive Learning (NCL), we follow the default SimCLR settings, with the addition of a non-negative constraint using the ReLU function. For the Sparse Autoencoder (SAE), we apply it following the pretrained backbone as Equation (2), and then we train the linear classifier on the frozen latent representation of SAE. More details can be found in Appendix A.1.

**Robustness Against Label Noise.** When evaluating the robustness against label noise, we train a linear classifier following the frozen pretrained encoders, where the labels are uniformly flipped to the other classes with a probability $\eta$ (noise rate). As shown in Table 2, when the linear classifiers are trained on the samples with clean labels, monosemantic and polysemantic features exhibit comparable performance. However, in the presence of label noise, both NCL and SAE significantly outperform across various datasets. Especially, when the noise rates are aggressive, the improvements are substantial, with NCL showing a 13.7% improvement on ImageNet-100 and a 9.2% improvement on CIFAR-100 under 90% noisy labels. The results are consistent with the results in toy models and further verify that monosemnatic features obtain stronger robustness against label noise.

**Robustness Against Distribution Shifts.** For evaluating the resilience of features to distribution shifts, we evaluate three types of shifts, including random input noise, random Gaussian noise, and real-world distribution shifts (Wang et al., 2019a; Geirhos et al., 2018; Hendrycks & Dietterich, 2019) on ImageNet-100 datasets. The models and classifiers are trained on the clean ImageNet-100 dataset while their classification performance is evaluated with noisy samples. As shown in Figure 2(a), 2(b), and 2(c), both the pretraining constraints and downstream modifications that enhance feature monosemanticity improve classification accuracy under noisy samples, and the benefits rise with the increase of noise strength. The results suggest that the monosemantic features can also enhance the robustness against various noises applied in inputs.

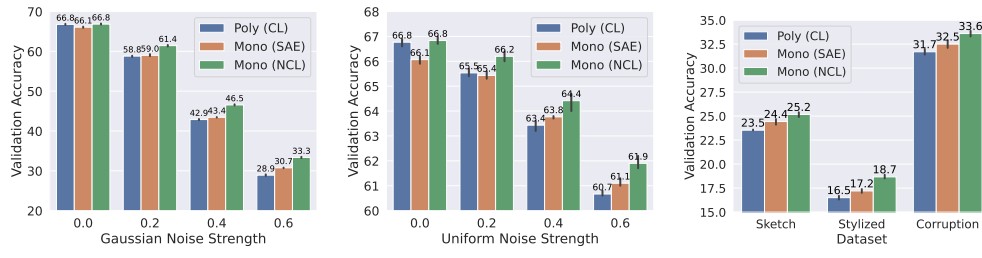

| (a) Gaussian Input Noise | (b) Uniform Input Noise | (c) Real-world Distribution Shift |

Figure 2: The evaluation of robustness against input distribution shifts on ImageNet-100. Monosemantic representations (SAE, NCL) exhibit improvements in the robustness against different kinds of distribution shifts.

## 3.2 MONOSEMANTIC FEATURES ARE ROBUST UNDER FEW-SHOT AND NOISY FINETUNING

In practice, fully finetuning a large pretrained model on downstream labeled data can often achieve better performance than linear probing, but it also easily overfits if there is only a few amount of labeled data. Here, we compare standard finetuning (polysemantic) to monosemantic finetuning.

### 3.2.1 METHODS FOR MONOSEMANTIC FINETUNING

**Standard Finetuning.** For the baseline, we consider a common finetuning setting, i.e., we pretrain the encoders with contrastive learning on unlabeled ImageNet-100 and then learn a linear classifier on labeled Imagenet-100 with the cross-entropy loss: $\mathcal{L}_{\text{CE}}(f) = \mathbb{E}_{x,y} \log \frac{\exp(f(x)^\top w_y)}{\sum_{c=1}^{C} \exp(f(x)^\top w_c)}$, where $f$ is the encoder network and $w_c$ is the linear classifier weight of the related label. Unlike linear probing, we train classifiers on the pretrained representations without clipping the gradient of encoders.

**Non-negative Tuning.** According to NCL (Wang et al., 2024), replacing the original cross-entropy (CE) loss used in the supervised learning process with the non-negative cross-entropy (NCE) loss will *maintain monosemanticity during supervised learning*. Thus, we use it as a monosemantic finetuning strategy. To be specific, NCE applies non-negative transformation to the representations $f(x)$, i.e.,

$$\mathcal{L}_{\text{NCE}}(f) = \mathbb{E}_{x,y} \log \frac{\exp(f_+(x)^\top w_y)}{\sum_{c=1}^{C} \exp(f_+(x)^\top w_c)}, \tag{3}$$

where $f_+(x) = \sigma(f(x))$ with a non-negative activation function, e.g., ReLU. By respectively finetuning contrastive pretrained models with CE and NCE objectives, we compare the robustness of polysemantic and monosemantic features across two different tasks: few-shot finetuning and noisy label finetuning.

**Evaluation of Monosemanticity.** We still measure the monosemanticity of different models by the semantic consistency (Wang et al., 2024). We evaluate the models in different cases, including models fine-tuned with label noise and few-shot samples. As shown in Table 1, non-negative tuning significantly improves the semantic consistency of trained models.

### 3.2.2 EXPERIMENTS

**Few-shot Finetuning.** As the finetuning process usually involves fewer training samples, a crucial challenge for feature robustness is preventing overfitting on small training datasets. To evaluate the performance of polysemantic and monosemantic features during few-shot finetuning, we respectively use 10%, 20%, 50% and the entire training set of ImageNet-100 to finetune the pretrained representations with CE and NCE objectives. As shown in Figure 3(a), 3(b), the monosemantic features exhibit **lower training accuracy but higher validation accuracy** in few-shot finetuning, and the advantages grow when the training set becomes smaller, which implies that the monosemanticity helps representations to be less likely to overfit the training set in the downstream task.

**Noisy-label Finetuning.** We also evaluate robustness against label noise in finetuning tasks on ImageNet-100. During the finetuning process, the labels of training samples are uniformly flipped to the other classes with a probability $\eta$ (noise rate). As shown in Figure 3(c), non-negative finetuning

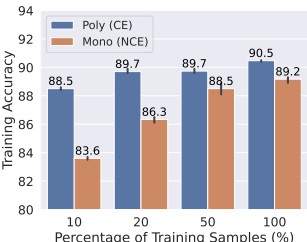 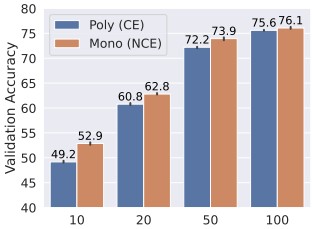 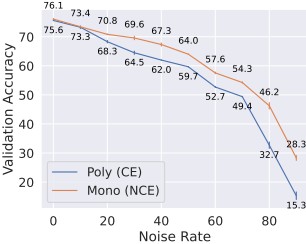

(a) Training Accuracy of Few-shot Finetuning   (b) Validation Accuracy of Few-shot Finetuning   (c) Noisy Label Finetuning

Figure 3: The robustness of the models finetuned with polysemanticity (CE) and monosemanticity (NCE) under different noises on ImageNet-100. Attaining monosemanticity during the finetuning process enhances the robustness across various tasks.

leads to significant gains under label noise that keep growing with the increase of the noise rate. Notably, monosemantic features exhibit **at most 11.9% improvement** under large noise rate.

These empirical results indicate that maintaining feature monosemanticity during the finetuning process can bring better learning robustness against overfitting and label noise.

### 3.3 MONOSEMANTIC LORA FOR LARGE LANGUAGE MODELS

When evaluating, we consider a common scenario related to robustness in large language model fine-tuning. Specifically, during the fine-tuning process, the large language models often compromise the already learned alignment, which leads to a security risk (Qi et al., 2023; Li et al., 2025). In practice, we use the Llama-2-7B-Chat (Touvron et al., 2023) as the aligned model and further finetune it with SST2 (Socher et al., 2013) as the review sentiment classification task and Dolly (Conover et al., 2023) datasets as the dialogue generation task. To evaluate the security and alignment performance, we use the ShieldGemma-9B (Zeng et al., 2024) and Beavertails-7B (Ji et al., 2024) models to evaluate the alignment of model responses based on the response on Beavertails datasets (Ji et al., 2024). More details of the parameters and the metrics can be found in Appendix A.3.

Table 3: Evaluation of LoRA and MonoLoRA with Llama-2-7B-Chat fine-tuned on SST2 and Dolly datasets. Task performance is measured by accuracy (SST2) and RougeL score (Dolly) (Lin, 2004). Alignment performance is assessed based on responses to the Beavertails dataset.

| Dataset | Model | ShieldGemma Alignment Scores ($\downarrow$) | | | | | Alignment Sparsity | Task Sparsity | Beavertails Scores ($\downarrow$) | Task Perf. ($\uparrow$) |
| | | Danger. | Harass. | Hate. | Sex. | Avg | | | | |
| SST2 | Base | 7.66 | 2.88 | 6.14 | 2.64 | 4.83 | - | - | 20.90 | 88.65 |
| | LoRA | 8.48 | 6.91 | 9.43 | 6.77 | 7.90 | 0 | 0 | 20.60 | 92.78 |
| | **MonoLoRA** | **5.37** | **2.23** | **4.63** | **1.88** | **3.53** | 45.54 | 36.71 | **20.00** | **94.84** |
| Dolly | Base | 7.66 | 2.88 | 6.14 | 2.64 | 4.83 | - | - | 20.90 | 10.21 |
| | LoRA | 10.54 | **3.53** | 7.53 | 2.86 | 6.12 | 0 | 0 | 23.80 | 14.08 |
| | **MonoLoRA** | **10.49** | 3.56 | **7.40** | **2.70** | **6.04** | 38.69 | 40.00 | **22.60** | **14.48** |

As shown in Table 3, MonoLoRA models exhibit greater alignment resilience compared to standard LoRA models while maintaining comparable fine-tuning performance. To assess monosemanticity, we adopt sparsity (zero-value ratio) of intermediate activations as a surrogate metric (Gurnee et al., 2023) and evaluate it on the Beavertails (Alignment Sparsity), SST2 (Task Sparsity), and Dolly (Task Sparsity) datasets. Higher sparsity indicates stronger monosemanticity. Our results suggest that enhancing monosemanticity at the neuron level improves the robustness of LLMs, mitigating the risk of compromising learned capabilities when fine-tuned with downstream data.

## 4 UNDERSTANDING THE ROBUSTNESS GAINS OF MONOSEMANTICITY

In Section 3, we provide a comprehensive evaluation of the robustness gains of feature monosemanticity across multiple scenarios. Yet, we do not have a fully clear understanding of *why monosemantic*

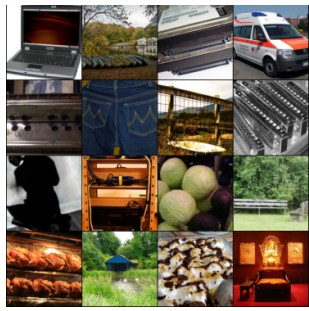

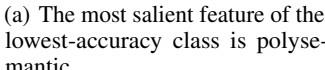

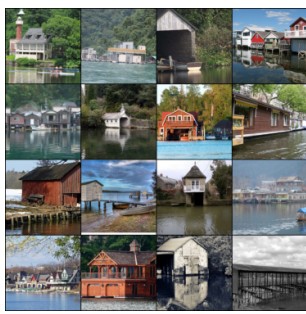

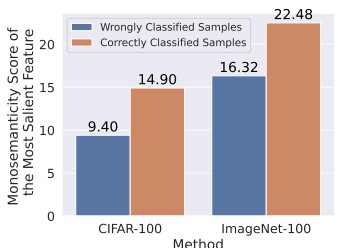

(a) The most salient feature of the lowest-accuracy class is polysemantic.

(b) The most salient feature of the highest-accuracy class is monosemantic.

(c) Correctly classified samples have more monosemantic features.

Figure 4: Influence of feature monosemanticity on classification performance, where the classifier is applied after a frozen contrastive encoder and trained with 90% noisy labels. (a), (b) respectively draw the activated samples on the dimensions with the largest clssifier weight of the lowest-accuracy and highest-accuracy classes on ImageNet-100. (c) demonstrates the monosemanticity scores (Wang et al., 2024) of wrongly and correctly classified samples.

*features are more robust.* As a preliminary step to demystify this phenomenon, in this section, we investigate the influence of monosemanticity on learned classifiers from both empirical (Section 4.1) and theoretical (Sections 4.2 & 4.3) perspectives. For simplicity, we focus on the label noise scenario.

## 4.1 NOISY CLASSIFIERS PREFER MONOSEMANTIC FEATURES IN PRACTICE

To further understand the robustness improvements brought by monosemanticity, we investigate the difference in the salient features of the robust and non-robust classifiers under noisy conditions. Taking the linear classifier trained on ImageNet-100 with 90% noisy labels (Section 3.1)) as an example, we start with respectively visualizing the dominant features for classes with the highest and lowest accuracy. For each class, we find the feature dimension with the largest classifier weight for the ground-truth label and visualize the top-activated samples along the dimension. As shown in Figure 4(a), 4(b), we observe a clear difference: samples activated in the dimension related to the lowest accuracy class (jeans) belong to different classes while samples activated in the dimension related to the highest accuracy class (boathouse) share the same label, i.e., the classifier with higher performance under label noise relies on a more monosemantic dimension.

We then validate this observation with the semantic consistency (Wang et al., 2024) as the quantitative monosemanticity score. The semantic consistency calculates the proportion of activated samples that belong to their most frequent class along a dimension. With a larger semantic consistency, the dimension is more likely to be activated by the samples from the same class, i.e., the feature is more monosemantic. To compare the robust and non-robust classifiers, we respectively draw the samples that are wrongly and correctly classified by the classifiers learned on ImageNet-100 with 90% noisy labels. For the embedding of each sample, we draw the dimension with the largest activation value and calculate the semantic consistency. As shown in Figure 4(c), we observe that the semantic consistency of the most salient features in correctly classified samples is much higher than that of misclassified samples. **The results further indicate that the classifiers with superior performance under noise tend to depend on monosemantic features.**

## 4.2 REPLICATING MONOSEMANTICITY GAINS WITH THE SUPERPOSITION MODEL

To further establish a theoretical understanding of the robustness benefits brought by monosemanticity, we introduce a toy model proposed by Elhage et al. (2022b) for the simplicity of analysis. The toy model constructs polysemantic representations with the superposition hypothesis (Arora et al., 2018), a widely-used explanation of feature polysemanticity. The hypothesis states that a polysemantic feature is an approximately linear combination of multiple latent semantics while a monosemantic feature is the reconstruction of a single natural concept. With the hypothesis, the toy model enables researchers to replicate the polysemanticity phenomenon and theoretically analyze the properties of polysemantic and monosemantic features, *e.g.,* occurrence conditions, learning dynamics, and

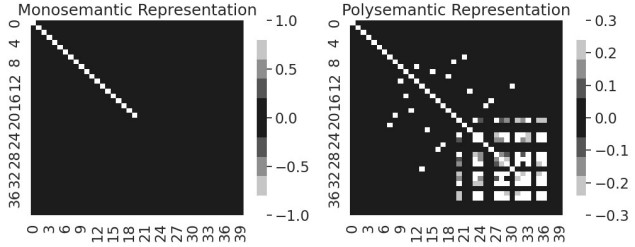
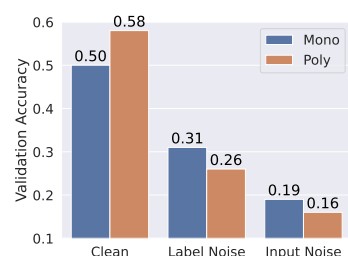

(a) Polysemantic dimensions correspond to multiple latent semantics.

(b) Polysemantic features have worse performance under noisy data.

Figure 5: The comparison between polysemantic and monosemantic features on the toy model introduced by Elhage et al. (2022b) ($n = 40$, $m = 20$, $S = 0.2$). (a) demonstrates the Parameters ($W^\top W$) of monosemantic (Left) and polysemantic features (Right) on the Toy Model. (b) evaluates the classification performance of features against different noises. The label noise denotes applying 90% noisy labels to the training samples and input noise denotes applying Gaussian noise to the validation samples.

geometric structures (Lecomte et al., 2024; Marshall & Kirchner, 2024; Chen et al., 2023). In this section, we start by introducing the setups and observing the robustness of different features on the toy model.

**Toy Model Setups.** In practice, we follow the settings proposed by Elhage et al. (2022b) and evaluate the robustness of polysemantic features on the toy model. Specifically, we assume each sample $x$ has $n$ dimensions and each dimension represents a natural concept. As the features in real-world datasets are usually sparsely activated (Olshausen & Field, 1997), we assume each dimension of a sample $x$ has an associated sparsity $S$ and let $x_i = 0$ with probability $S$. If not zero, we let each dimension be uniformly distributed between $[0, 1]$. When evaluating the performance, we consider the classification tasks of natural concepts, i.e., the labels satisfy that $y(x) = \arg\max_i x_i$. For the encoding network, we consider a linear model $h = W^\top W x$, where $W \in \mathbb{R}^{m \times n}$, with $m < n$, i.e., the hidden dimension is smaller than the input dimension. In practice, we use the reconstruction of $x$ as the training objective and obtain two kinds of learned features. As shown in Figure 5(a), when the superposition does not occur, we observe that $W^\top W$ is diagonal and has only $m$ non-zero elements, which means the model only captures $m$ concepts and each dimension is monosemantic. In contrast, when superposition happens, features obtain more concepts than the model dimensions and different concepts are projected into the same dimension.

**Noisy Learning Settings.** To evaluate the robustness of features, we respectively add noise to the labels and samples. For training with noisy labels, we denote the noise rate as $\eta$, where each label $y$ is uniformly switched to one of the other $n - 1$ labels with probability $\eta$. In experiments, we selected an aggressive noise rate (90%). With labeled samples, we train a linear classifier following the frozen features and evaluate the classification accuracy on a validation dataset without noisy labels. For noisy sample validation, we train a linear classifier on the clean dataset and add the Gaussian noise to the validation set samples.

**Empirical Results.** As shown in Figure 5(b), in the absence of noise, polysemantic features exhibit better performance, which is expected as the superposition enables features to capture more concepts. However, when there exists noise in the labels and samples, the situation changes significantly. The feature without superposition shows improvements over that with superposition under both label noise and input noise. The empirical results replicate the phenomenon where the monosemantic features are more robust than polysemantic features.

### 4.3 THEORETICAL ANALYSES WITH THE SUPERPOSITION MODEL

After replicating the robustness gains of monosemanticity on the toy model, we then establish a theoretical comparison between polysemantic and monosemantic features. For ease of theoretical analysis, we consider a binary classification case in the toy model ($n = 2$, $m = 1$, $S = 0.2$). To be specific, a sample $x$ has two latent features $x_1, x_2$, and the model parameter $W \in \mathbb{R}^{1 \times 2}$. When we obtain the monosemantic features, the model output is $\nu_{\text{mono}} := x_1$. In contrast, when obtaining

polysemantic features, the model keeps more natural concepts than the representation dimension. According to Elhage et al. (2022b), one common geometric structure of polysemantic features is antipodal pairs formed by two concepts. Therefore, we assume the learned polysemantic feature to be $\nu_{\text{poly}} := x_1 - x_2$.

For conciseness of expressions, we introduce the following notations on mean and variance for a given feature representation $\nu$. For a clean distribution without label noise, we denote the conditional means and variances by $\mu_i(\nu) := \mathbb{E}(\nu|y = i)$, $\sigma_i^2(\nu) := \mathbb{E}((\nu - \mu_0(\nu))^2|y = i)$, $i = 0, 1$. For distinction, for a noisy distribution, we use $\tilde{\mu}$ and $\tilde{\sigma}$. Borrowing the concept from linear discriminant analysis (LDA) (Fisher, 1936), we deem that a good linearly discriminative representation should have a large distance between different classes whereas maintaining the intra-class variance as small as possible, i.e. maximize $\Delta\mu(\nu) = |\mu_0(\nu) - \mu_1(\nu)|$ whereas minimizing $\sigma_0^2(\nu)$ and $\sigma_1^2(\nu)$. Therefore, to quantitatively compare polysemantic and monosemantic representations, we use the criterion $J(\nu) = \Delta\mu(\nu)/(\sigma_0(\nu)\sigma_1(\nu))$. A larger value of $J(\nu)$ indicates better linear separability.

**Theorem 4.1** (Conditional means and variances of monosemantic & polysemantic features). *Let $\nu_{\text{mono}} = x_1$ and $\nu_{\text{poly}} = x_1 - x_2$. For conditional means, we have $\mu_0(\nu_{\text{poly}}) < \mu_0(\nu_{\text{mono}})$ and $\mu_1(\nu_{\text{poly}}) < \mu_1(\nu_{\text{mono}})$, yet $\Delta\mu(\nu_{\text{poly}}) > \Delta\mu(\nu_{\text{mono}})$. For conditional variances, we have $\sigma_1^2(\nu_{\text{poly}}) = \sigma_1^2(\nu_{\text{mono}})$ and $\sigma_0^2(\nu_{\text{poly}}) > \sigma_0^2(\nu_{\text{mono}})$. Overall, we have $J(\nu_{\text{poly}}) > J(\nu_{\text{mono}})$.*

According to the LDA criterion, the polysemantic feature with a larger $J(\nu)$ is more linearly separable. Intuitively, because the polysemantic embedding encodes information of both $x_1$ and $x_2$, it can do better classification w.r.t. the labels depending on both features. However, when there exits label noise, we observe a different situation.

**Theorem 4.2** (Influence of label noise on linear seprarability). *We denote the linear separability criterion under noise as $\tilde{J}(\nu) = \Delta\tilde{\mu}(\nu)/(\tilde{\sigma_0}(\nu)\tilde{\sigma_1}(\nu))$. For noise rate $\eta \in [0, 0.5)$,*

$$\frac{\tilde{J}(\nu_{\text{poly}})}{J(\nu_{\text{poly}})} \leq \frac{\tilde{J}(\nu_{\text{mono}})}{J(\nu_{\text{mono}})} \leq 1. \tag{4}$$

*Meanwhile, we obtain $\tilde{J}(\nu_{\text{poly}}) \leq \tilde{J}(\nu_{\text{mono}})$ when $\eta \in [0.25, 0.5)$.*

As shown in Theorem 4.2, with the increase of noise rate, the linear separability ($J(\nu)$) of both polysemantic and monosemantic features becomes worse. However, $J(\nu_{mono})$ decreases more slowly. As a result, when the noise rate is aggressive enough ($\eta \geq 0.25$), the monosemantic feature exhibits better linear seperability than the polysemantic one. Moreover, in Appendix B.3, we show that input noise has a similar influence on linear separability. The theoretical results reveal that the linear separability of monosemantic features is more robust than polysemantic ones in the toy model settings, which leads to better performance in tasks under noise. As a preliminary step, we believe that the examples can inspire further theoretical analysis of the relationship between monosemanticity and model robustness in more realistic situations, such as fine-tuning scenarios.

## 5 CONCLUDING REMARKS

Recent work has made significant strides in enhancing model interpretability by promoting feature monosemanticity through various techniques. However, a prevailing belief in the literature posits an accuracy-interpretability tradeoff, suggesting that achieving monosemantic features for better interpretability necessarily compromises prediction accuracy. In this study, we have challenged this notion by demonstrating the advantages of monosemanticity beyond interpretability alone. Specifically, we found that monosemantic features are significantly more robust to various types of distribution shifts, including input noise, label noise, and real-world out-of-domain inputs. Additionally, we have shown that maintaining feature monosemanticity during fine-tuning serves as an effective regularizer, reducing model overfitting in few-shot settings, noisy environments, and during large language model (LLM) fine-tuning. We also provide an in-depth analysis of the benefits of monosemantic features from both theoretical and empirical aspects. These diverse sources of learning robustness collectively indicate that monosemantic features have a general sense of robustness, resonating with its benefits in interpretability. Therefore, rather than viewing monosemanticity as a necessary cost for interpretability, we advocate for embracing and exploring the multiple learning advantages it offers. We believe our work, as a pioneering effort in this direction, will inspire future research to investigate these possibilities further.

## REPRODUCIBILITY STATEMENT

To ensure the reproducibility of our results, we elaborate on the details of our experiments and theoretical analysis in the main paper and the appendix. In Section 3.1, 3.2, 3.3 of the main paper, we respectively introduce the methods for capturing polysemantic and monosemantic features in linear probing, finetuning vision models and finetuning LLMs. Furthermore, in Appendix A, we introduce the hyperparameters and implementation details of adopted methods, and the detailed settings of the robustness evaluation, including input and label noise, few-shot learning, and out-of-domain generalization. For theoretical results, we introduce the toy models in Section 4.3 of the main paper and provide detailed proofs and explanations for the theoretical comparison in Appendix B.

## ACKNOWLEDGEMENT

Yisen Wang was supported by National Key R&D Program of China (2022ZD0160300), National Natural Science Foundation of China (92370129, 62376010), and Beijing Nova Program (20230484344, 20240484642). Yifei Wang and Stefanie Jegelka were supported in part by the NSF AI Institute TILOS, and an Alexander von Humboldt Professorship.

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

# A    EXPERIMENT DETAILS

## A.1    EXPERIMENT DETAILS FOR NOISY LINEAR PROBING

During the pretraining process, we utilize ResNet-18 (He et al., 2016) as the backbone and train the models on CIFAR-100 and ImageNet-100. We pretrain the model for 200 epochs. The projector is a two-layer MLP with a hidden dimension 16384 and an output dimension 2048. We train the models with batch size 256 and weight decay 0.0001. When implementing NCL and SAE, we follow the default settings of SimCLR. For NCL, we adopt ReLU as the activation function $\sigma$. For SAE, the encoder and decoder are linear layers with 2048 input and output dimensions, and the number of activated features in the hidden layer is 256.

During the linear evaluation, we train a classifier following the frozen backbone pretrained by different methods for 50 epochs. For noisy label probing, we apply symmetric label noise when training the linear classifiers, i.e., the labels are uniformly flipped to the other classes with the noisy rate. And for random input noise, we train the linear classifiers on clean datasets, while applying different scales of uniform noise and Gaussian noise to the validation sample. For real-world out-of-domain distribution shifts, we use ImageNet-sketch and ImageNet-stylized datasets (Geirhos et al., 2018; Wang et al., 2019a). As we pretrain the network on ImageNet-100, we select the samples of the corresponding 100 classes from these out-of-distribution datasets and evaluate the accuracy.

## A.2    EXPERIMENT DETAILS FOR FEW-SHOT AND NOISY FINETUNING FROM PRERTRAINED FEATURES

During the pretraining process, we utilize ResNet-18 (He et al., 2016) as the backbone and train the models on ImageNet-100. We pretrain the model for 200 epochs. We use a projector which is a two-layer MLP with hidden dimension 16384 and output dimension 2048. We pretrain the models with batch size 256 and weight decay 0.0001. During the finetuning process, we train a classifier following the backbone for 100 epochs respectively with standard and non-negative tuning, following the default settings of finetuning. When implementing the non-negative tuning, we select the ReLU function as the non-negative operator. For few-shot finetuning, we respectively random draw 10%, 20%, 50%, and 100$ training samples from the original ImageNet-100 training set. For noisy label fine-tuning, we still apply symmetric label noise with different noise rates to the training samples.

## A.3    EXPERIMENT DETAILS FOR MONOSEMANTIC LLM FINETUNING

**Hyper-parameters** We finetune the Llama-2-7b-Chat model in SST2 with 20 epochs, batch size 16 and learning rate 1e-4. we Lora with rank $r = 8$, scaling factor $\alpha = 4$, and dropout rate 0.1 as default. For Dolly, we finetune it 1 epoch (by common practice), with batch size 4. The LoRA module is added to every query and value mapping module in the base model. For finetuning we use 5000 samples from the dataset. For inference, we use 1000 samples.

**Security Scores**: We use the ShieldGemma-9B (Zeng et al., 2024) and Beavertails-7B (Ji et al., 2024) models to evaluate the alignment of model responses based on the response on Beavertails datasets (Ji et al., 2024). The Bevertails-7B and ShieldGemma-9B are sheild models which can detect the unsafe behaviors of models.

**Prompt Template**

Listing 1: SST-2 Prompt

```
Below is an instruction that describes a task.
Write a response that appropriately completes the request.

### Instruction:
Analyze the sentiment of the input, and respond only 'positive' or 'negative'.

### Input:
{sentence}

### Response:
```

Listing 2: ShieldGemma Guideline

```
\"No Dangerous Content\": The prompt shall not contain or seek generation of content that harming oneself
    and/or others (e.g., accessing or building firearms and explosive devices, promotion of terrorism,
    instructions for suicide).
\"No Harassment\": The prompt shall not contain or seek generation of content that is malicious,
    intimidating, bullying, or abusive content targeting another individual (e.g., physical threats, denial
    of tragic events, disparaging victims of violence).
\"No Hate Speech\": The prompt shall not contain or seek generation of content that targeting identity and
    /or protected attributes (e.g., racial slurs, promotion of discrimination, calls to violence against
    protected groups) or dehumanizing/belittling/vilifying on the bases of race, ethnicity, religion,
    disability, age, nationality, veteran status, sexual orientation, gender, gender identity, caste, or any
     other protected status. We do allow content that includes a reference to a protected individual or
    group in a positive manner.
\"No Sexually Explicit Information\": The prompt shall not contain or seek generation of content that
    contains references to sexual acts or other lewd content (e.g., sexually graphic descriptions, content
    aimed at causing arousal). Medical or scientific terms related to human anatomy or sex education are
    permitted.
```

## A.4 EXPERIMENTS DETAILS FOR FIGURE 1

For Figure 1(a), we respectively draw a random dimension from the models trained by CL and NCL, and then draw the top-activated samples along two dimensions. We utilize ResNet-18 (He et al., 2016) as the backbone and train the models on ImageNet-100 for 200 epochs.

For Figure 1(b), we evaluate the performance in linear probing with noise. During the linear evaluation, we train a classifier following the frozen backbone pretrained by different methods for 50 epochs. For noisy label probing, we apply 90% symmetric label noise when training the linear classifiers. For random input noise, we train the linear classifiers on clean datasets, while applying Gaussian noise with 0.6 standard variation to the validation sample.

## B PROOFS

### B.1 PROOFS RELATED TO THEOREM 4.1

#### B.1.1 MONOSEMANTIC REPRESENTATIONS

In the monosemantic case, we assume the learned representation only keeps the most important dimension $\nu = x_1$.

**Theorem B.1** (Conditional mean and variance of monosemantic representations). *The conditional means and variances of $\nu_{\mathrm{mono}} = x_1$ are*

$$\mu_0(\nu_{\mathrm{mono}}) = \frac{1}{3}\frac{(1-S)^2}{1+S^2} \quad and \quad \mu_1(\nu_{\mathrm{mono}}) = \frac{1}{3}\frac{2+S}{1+S} \tag{5}$$

$$\sigma_0^2(\nu_{\mathrm{mono}}) = \frac{1}{6}\frac{(1-S)^2}{1+S^2} - \mu_0(\nu_{\mathrm{mono}})^2 \quad and \quad \sigma_1^2(\nu_{\mathrm{mono}}) = \frac{1}{6}\frac{3+S}{1+S} - \mu_1(\nu_{\mathrm{mono}})^2. \tag{6}$$

*Proof of Theorem B.1.* We first calculate the conditional probability density functions.

$$
\begin{aligned}
&\mathrm{P}(x_1 \leq x | y = 0)\\
&= \frac{\mathrm{P}(x_1 \leq x, x_1 \leq x_2)}{\mathrm{P}(x_1 \leq x_2)}\\
&= \frac{\mathrm{P}(x_1 \leq x_2, x_1 \leq x, x_2 \leq x) + \mathrm{P}(x_1 \leq x_2, x_1 \leq x, x_2 > x)}{\mathrm{P}(x_1 \leq x_2)}\\
&= \frac{\mathrm{P}(x_1 = 0, x_2 \leq x) + \mathrm{P}(x_1 \leq x_2, 0 < x_1 \leq x, 0 < x_2 \leq x) + \mathrm{P}(x_1 \leq x)\mathrm{P}(x_2 > x)}{\mathrm{P}(x_1 \leq x_2)}\\
&= \frac{S[S + (1-S)x] + \frac{1}{2}(1-S)^2 x^2 + [S + (1-S)x](1-S)(1-x)}{\frac{1}{2}(1+S^2)}\\
&= 1 - \frac{(1-S)^2(1-x)^2}{1+S^2}.
\end{aligned} \tag{7}
$$

$$
\begin{aligned}
\mathrm{P}(x_1 \le x | y = 1) &= \frac{\mathrm{P}(x_1 \le x, x_1 > x_2)}{\mathrm{P}(x_1 > x_2)} \\
&= \frac{\mathrm{P}(x_1 \le x) - \mathrm{P}(x_1 \le x, x_1 \le x_2)}{\mathrm{P}(x_1 > x_2)} \\
&= \frac{\mathrm{P}(x_1 \le x)}{\mathrm{P}(x_1 > x_2)} - \mathrm{P}(x_1 \le x | y = 0) \cdot \frac{\mathrm{P}(x_1 \le x_2)}{\mathrm{P}(x_1 > x_2)} \\
&= \frac{S + (1-S)x}{\frac{1}{2}(1 - S^2)} - \left[ 1 - \frac{(1-S)^2(1-x)^2}{1 + S^2} \right] \cdot \frac{1 + S^2}{1 - S^2} \\
&= \frac{(1-S)^2 x^2 + 2S(1-S)x}{1 - S^2}.
\end{aligned}
\tag{8}
$$

Then the conditional means of $\nu_{\mathrm{mono}} = x_1$ are

$$
\begin{aligned}
\mu_0(\nu_{\mathrm{mono}}) &= \int_x x \, d\mathrm{P}(x_1 \le x | y = 0) \\
&= \int_{x \in (0,1]} x \frac{2(1-S)^2}{1 + S^2}(1-x) \, dx \\
&= \frac{2(1-S)^2}{1 + S^2} \left( \frac{1}{2} - \frac{1}{3} \right) \\
&= \frac{1}{3} \frac{(1-S)^2}{1 + S^2}
\end{aligned}
\tag{9}
$$

and

$$
\begin{aligned}
\mu_1(\nu_{\mathrm{mono}}) &= \int_x x \, d\mathrm{P}(x_1 \le x | y = 1) \\
&= \int_{x \in (0,1]} x \cdot 2(1-S) \frac{(1-S)x + S}{1 - S^2} \, dx \\
&= \frac{2(1-S)}{1 - S^2} \left[ \frac{1}{3}(1-S) + \frac{1}{2}S \right] \\
&= \frac{1}{3} \frac{2 + S}{1 + S}.
\end{aligned}
\tag{10}
$$

Then we have

$$
\mu_1(\nu_{\mathrm{mono}}) - \mu_0(\nu_{\mathrm{mono}}) = \frac{1}{3} \frac{1 + S}{1 + S^2}.
\tag{11}
$$

Similarly, we have the conditional variances as follows.

$$
\begin{aligned}
\sigma_0^2(\nu_{\mathrm{mono}}) &= \int_x x^2 d\mathrm{P}(x_1 \le x | y = 0) - \mu_0(\nu_{\mathrm{mono}})^2 \\
&= \int_{x \in (0,1]} x^2 \frac{2(1-S)^2}{1 + S^2}(1-x) \, dx - \mu_0(\nu_{\mathrm{mono}})^2 \\
&= \frac{2(1-S)^2}{1 + S^2} \left[ \frac{1}{3} - \frac{1}{4} \right] - \mu_0(\nu_{\mathrm{mono}})^2 \\
&= \frac{1}{6} \frac{(1-S)^2}{1 + S^2} - \mu_0(\nu_{\mathrm{mono}})^2
\end{aligned}
\tag{12}
$$

and

$$
\begin{aligned}
\sigma_1^2(\nu_{\mathrm{mono}}) &= \int_x x \, d\mathrm{P}(x_1 \le x | y = 1) - \mu_1(\nu_{\mathrm{mono}})^2 \\
&= \int_{x \in (0,1]} x^2 \cdot 2(1-S) \frac{(1-S)x + S}{1 - S^2} \, dx - \mu_1(\nu_{\mathrm{mono}})^2 \\
&= \frac{2(1-S)}{(1-S^2)} \left[ \frac{1}{4}(1-S) + \frac{1}{3}S \right] - \mu_1(\nu_{\mathrm{mono}})^2 \\
&= \frac{1}{6} \frac{3+S}{1+S} - \mu_1(\nu_{\mathrm{mono}})^2.
\end{aligned}
\tag{13}
$$

$\square$

### B.1.2 Polysemantic Representations

To study the polysemantic case, we first have to derive the probability distribution of $\nu_{\mathrm{poly}} = x_1 - x_2$ and the corresponding conditional probability density functions on $y = 0$ and $y = 1$, separately. We first calculate the cumulative distribution functions as follows.

**Lemma B.2** (Distribution of $\nu_{\mathrm{poly}} = x_1 - x_2$)**.**

$$
\mathrm{P}(x_1 - x_2 \le x) = \begin{cases}
-\dfrac{1}{2}[1 - (1-S)x]^2 + 1 + \dfrac{1}{2}S^2, & x \in [0, 1], \\
\dfrac{1}{2}[(1-S)x + 1]^2 - \dfrac{1}{2}S^2, & x \in [-1, 0).
\end{cases}
$$

*Proof of Lemma B.2.* For $x \in [0, 1]$, we have

$$
\begin{aligned}
&\mathrm{P}(x_1 - x_2 \le x) \\
&= \lim_{N \to \infty} \sum_{n=-N}^{N} \mathrm{P}(x_1 \le x + n/N)\mathrm{P}(x_2 = n/N) \\
&= \lim_{N \to \infty} \sum_{n=0}^{\lfloor (1-x)N \rfloor} \mathrm{P}(x_1 \le x + n/N)\mathrm{P}(x_2 = n/N) + \sum_{n=\lfloor (1-x)N \rfloor + 1}^{N} 1 \cdot \mathrm{P}(x_2 = n/N) \\
&= [S + (1-S)x] \cdot S \\
&\quad + \lim_{N \to \infty} \sum_{n=1}^{\lfloor (1-x)N \rfloor} [S + (1-S)(x + n/N)] \cdot (1-S)/N + \sum_{n=\lfloor (1-x)N \rfloor + 1}^{N} (1-S)/N \\
&= S[S + (1-S)x] + \lim_{N \to \infty} [S(1-S) + (1-S)^2 x]\lfloor (1-x)N \rfloor / N \\
&\quad + (1-S)^2 \lfloor (1-x)N \rfloor (\lfloor (1-x)N \rfloor + 1)/(2N^2) + (1-S)(N - \lfloor (1-x)N \rfloor - 1)/N \\
&= S[S + (1-S)x] + [S(1-S) + (1-S)^2 x](1-x) + (1-S)^2(1-x)^2/2 + (1-S)x \\
&= -\frac{1}{2}[1 - (1-S)x]^2 + 1 + \frac{1}{2}S^2.
\end{aligned}
\tag{14}
$$

For $x \in [-1, 0)$, we have

$$
\begin{aligned}
\mathrm{P}(x_1 - x_2 \leq x) &= \lim_{N \to \infty} \sum_{n=-N}^{N} \mathrm{P}(x_1 \leq x + n/N)\mathrm{P}(x_2 = n/N) \\
&= \lim_{N \to \infty} \sum_{n=-\lfloor xN \rfloor}^{N} \mathrm{P}(x_1 \leq x + n/N)\mathrm{P}(x_2 = n/N) \\
&= \lim_{N \to \infty} \sum_{n=-\lfloor xN \rfloor}^{N} [S + (1-S)(x + n/N)] \cdot (1-S)/N \\
&= \lim_{N \to \infty} [S(1-S) + (1-S)^2 x](N + \lfloor xN \rfloor)/N \\
&\quad + (1-S)^2(N - \lfloor xN \rfloor)(N + \lfloor xN \rfloor + 1)/(2N^2) \\
&= [S(1-S) + (1-S)^2 x](1+x) + (1-S)^2(1-x^2)/2 \\
&= \frac{1}{2}[(1-S)x + 1]^2 - \frac{1}{2}S^2.
\end{aligned}
\tag{15}
$$

$\square$

**Theorem B.3** (Conditional mean and variance of polysemantic representations). *The conditional means and variances of $\nu_{\mathrm{poly}} = x_1 - x_2$ are*

$$
\mu_0(\nu_{\mathrm{poly}}) = -\frac{1}{3}\frac{(1-S)(1+2S)}{1+S^2} \quad and \quad \mu_1(\nu_{\mathrm{mono}}) = \frac{1}{3}\frac{1+2S}{1+S}
\tag{16}
$$

$$
\sigma_0^2(\nu_{\mathrm{poly}}) = \frac{1}{6}\frac{(1-S)(1+3S)}{1+S^2} - \mu_0(\nu_{\mathrm{poly}})^2 \quad and \quad \sigma_1^2(\nu_{\mathrm{poly}}) = \frac{1}{6}\frac{1+3S}{1+S} - \mu_1(\nu_{\mathrm{mono}})^2
\tag{17}
$$

*Proof of Theorem B.3.* By Lemma B.2, we have

$$
\begin{aligned}
\mathrm{P}_{\mathrm{poly}}(x_1 - x_2 \leq x | y = 0) &= \mathrm{P}(x_1 - x_2 \leq x | x_1 \leq x_2) \\
&= \mathrm{P}(x_1 - x_2 \leq \min(0, x))/\mathrm{P}(x_1 - x_2 \leq 0) \\
&= \begin{cases} \left[\frac{1}{2}[(1-S)x+1]^2 - \frac{1}{2}S^2\right]/[\frac{1}{2}(1+S^2)], & x \in [-1, 0) \\ 1, & x \in [0, 1] \end{cases} \\
&= \begin{cases} \left[[(1-S)x+1]^2 - S^2\right]/(1+S^2), & x \in [-1, 0) \\ 1, & x \in [0, 1] \end{cases}
\end{aligned}
$$

and

$$
\begin{aligned}
&\mathrm{P}_{\mathrm{poly}}(x_1 - x_2 \leq x | y = 1) \\
&= \mathrm{P}(x_1 - x_2 \leq x | x_1 > x_2) \\
&= \mathrm{P}(0 < x_1 - x_2 \leq x)/\mathrm{P}(x_1 - x_2 > 0) \\
&= \begin{cases} 0, & x \in [-1, 0] \\ [\mathrm{P}(x_1 - x_2 \leq x) - \mathrm{P}(x_1 - x_2 \leq 0)]/[1 - \mathrm{P}(x_1 - x_2 \leq 0)], & x \in (0, 1] \end{cases} \\
&= \begin{cases} 0, & x \in [-1, 0] \\ \left[-\frac{1}{2}[1-(1-S)x]^2 + 1 + \frac{1}{2}S^2 - \frac{1}{2}(1+S^2)\right]/[1 - \frac{1}{2}(1+S^2)], & x \in (0, 1] \end{cases} \\
&= \begin{cases} 0, & x \in [-1, 0] \\ \left[1 - [1-(1-S)x]^2\right]/(1-S^2), & x \in (0, 1] \end{cases}
\end{aligned}
$$

Then we have

$$
\begin{aligned}
\mu_0(\nu_{\text{poly}}) &= \int_{x\in[-1,0)} x \cdot 2(1-S)[(1-S)x+1]/(1+S^2)\, dx \\
&= \frac{2(1-S)}{1+S^2}\left[\frac{1}{3}(1-S)-\frac{1}{2}\right] \\
&= -\frac{1}{3}\frac{(1-S)(1+2S)}{1+S^2},
\end{aligned}
\tag{18}
$$

$$
\begin{aligned}
\mu_1(\nu_{\text{poly}}) &= \int_{x\in(0,1]} x \cdot 2(1-S)[1-(1-S)x]/(1-S^2)\, dx \\
&= \frac{2}{1+S}\left[\frac{1}{2}-\frac{1}{3}(1-S)\right] \\
&= \frac{1}{3}\frac{1+2S}{1+S},
\end{aligned}
\tag{19}
$$

$$
\mu_1(\nu_{\text{poly}}) - \mu_0(\nu_{\text{poly}}) = \frac{2}{3}\frac{1+2S}{(1+S)(1+S^2)},
\tag{20}
$$

$$
\begin{aligned}
\sigma_0^2(\nu_{\text{poly}}) &= \int_{x\in[-1,0)} x^2 \cdot 2(1-S)[(1-S)x+1]/(1+S^2)\, dx - \mu_0(\nu_{\text{poly}})^2 \\
&= \frac{2(1-S)}{1+S^2}\left[-\frac{1}{4}(1-S)+\frac{1}{3}\right] - \mu_0(\nu_{\text{poly}})^2 \\
&= \frac{1}{6}\frac{(1-S)(1+3S)}{1+S^2} - \mu_0(\nu_{\text{poly}})^2,
\end{aligned}
\tag{21}
$$

and

$$
\begin{aligned}
\sigma_1^2(\nu_{\text{poly}}) &= \int_{x\in(0,1]} x^2 \cdot 2(1-S)[1-(1-S)x]/(1-S^2)\, dx - \mu_1(\nu_{\text{poly}})^2 \\
&= \frac{2}{1+S}\left[\frac{1}{3}-\frac{1}{4}(1-S)\right] - \mu_1(\nu_{\text{poly}})^2 \\
&= \frac{1}{6}\frac{1+3S}{1+S} - \mu_1(\nu_{\text{poly}})^2.
\end{aligned}
\tag{22}
$$

$\square$

### B.1.3 PROOF OF THEOREM 4.1

*Proof of Theorem 4.1.* Following the toy model described in Section 4.2, we let $S = 0.2$. Then by Theorem B.1, we have $\mu_0(\nu_{\text{mono}}) = 0.205$, $\mu_1(\nu_{\text{mono}}) = 0.611$, $\Delta\mu(\nu_{\text{mono}}) = 0.406$, $\sigma_0(\nu_{\text{mono}}) = 0.246$, $\sigma_1(\nu_{\text{mono}}) = 0.266$, and $J(\nu_{\text{mono}}) = 6.196$. By Theorem B.3, we have $\mu_0(\nu_{\text{poly}}) = -0.359$, $\mu_1(\nu_{\text{poly}}) = 0.389$, $\Delta\mu(\nu_{\text{poly}}) = 0.748$, $\sigma_0(\nu_{\text{mono}}) = 0.276$, $\sigma_1(\nu_{\text{poly}}) = 0.266$, and $J(\nu_{\text{poly}}) = 10.164$. By comparing the above results, we complete the proof. $\square$

### B.2 PROOFS RELATED TO LABEL NOISE

Following Ghosh et al. (2017); Ma et al. (2020); Wang et al. (2019b), we assume the noisy label $\tilde{y}$ is randomly flipped from the true labels to other classes. Under $\eta \in [0, \frac{K-1}{K})$, the noisy label distribution is

$$
P(\tilde{y}=k|x) = \sum_{j=0,1}^{K} P(\tilde{y}=k|y=j)P(y=k|x),
\tag{23}
$$

where $P(\tilde{y}=k|y=j) = 1-\eta$ if $j = k$, and otherwise $P(\tilde{y}=k|y=j) = \eta$.

### B.2.1 Influence of Label Noise on Conditional Mean and Variance

**Lemma B.4** (Conditional Distributions). *For noise rate $\eta \in [0, 1/2)$ and sparsity $S \in [0, 1]$, we have conditional distributions*

$$P(\nu|\tilde{y} = 0) = \frac{(1-\eta)(1+S^2)P(\nu|y = 0) + \eta(1-S^2)P(\nu|y = 1)}{(1-\eta)(1+S^2) + \eta(1-S^2)}, \tag{24}$$

*and*

$$P(\nu|\tilde{y} = 1) = \frac{\eta(1+S^2)P(\nu|y = 0) + (1-\eta)(1-S^2)P(\nu|y = 1)}{\eta(1+S^2) + (1-\eta)(1-S^2)}. \tag{25}$$

*Proof of Lemma B.4.* We first calculate the class conditional distributions.

$$\begin{aligned}
P(\nu|\tilde{y} = 0) &= P(\tilde{y} = 0|\nu)P(\nu)/P(\tilde{y} = 0) \\
&= \frac{\sum_{j=0,1} P(\tilde{y} = 0|y = j)P(y = j|\nu)P(\nu)}{\sum_{j=0,1} P(\tilde{y} = 0|y = j)P(y = j)} \\
&= \frac{\sum_{j=0,1} P(\tilde{y} = 0|y = j)P(\nu|y = j)P(y = j)}{\sum_{j=0,1} P(\tilde{y} = 0|y = j)P(y = j)} \\
&= \frac{(1-\eta)P(\nu|y = 0)P(y = 0) + \eta P(\nu|y = 1)P(y = 1)}{(1-\eta)P(y = 0) + \eta P(y = 1)}.
\end{aligned} \tag{26}$$

$$\begin{aligned}
P(\nu|\tilde{y} = 1) &= P(\tilde{y} = 1|\nu)P(\nu)/P(\tilde{y} = 1) \\
&= \frac{\sum_{j=0,1} P(\tilde{y} = 1|y = j)P(y = j|\nu)P(\nu)}{\sum_{j=0,1} P(\tilde{y} = 1|y = j)P(y = j)} \\
&= \frac{\sum_{j=0,1} P(\tilde{y} = 1|y = j)P(\nu|y = j)P(y = j)}{\sum_{j=0,1} P(\tilde{y} = 1|y = j)P(y = j)} \\
&= \frac{\eta P(\nu|y = 0)P(y = 0) + (1-\eta)P(\nu|y = 1)P(y = 1)}{\eta P(y = 0) + (1-\eta)P(y = 1)}.
\end{aligned} \tag{27}$$

Recall that $x_1, x_2 = 0$ with probability $S$, and $x_1, x_2 \sim \mathcal{U}(0, 1]$ with probability $1 - S$. Because $x_1$ and $x_2$ are independently and identically distributed and $P(x_1 = x_2|x_1, x_2 = 0)$, we have $P(x_1 \leq x_2|x_1, x_2 > 0) = P(x_2 \leq x_1|x_1, x_2 > 0) = 1/2$, and therefore

$$\begin{aligned}
\mathbb{P}(y = 0) &= \mathbb{P}(x_1 \leq x_2) \\
&= P(x_1 = 0) + P(x_1 > 0)P(x_2 > 0)P(x_1 \leq x_2|x_1, x_2 > 0) \\
&= S + \frac{1}{2}(1-S)^2 = \frac{1}{2}(1+S^2).
\end{aligned} \tag{28}$$

Then $P(y = 1) = 1 - P(y = 0) = \frac{1}{2}(1 - S^2)$, and correspondingly we have

$$P(\nu|\tilde{y} = 0) = \frac{(1-\eta)(1+S^2)P(\nu|y = 0) + \eta(1-S^2)P(\nu|y = 1)}{(1-\eta)(1+S^2) + \eta(1-S^2)}, \tag{29}$$

and

$$P(\nu|\tilde{y} = 1) = \frac{\eta(1+S^2)P(\nu|y = 0) + (1-\eta)(1-S^2)P(\nu|y = 1)}{\eta(1+S^2) + (1-\eta)(1-S^2)}. \tag{30}$$

$\square$

**Theorem B.5** (Influence of label noise on inter-class distance). *For noise rate $\eta \in [0, \frac{1}{2})$,*

$$\Delta\tilde{\mu}(\nu) = \frac{(1-2\eta)(1+S^2)(1-S^2)}{[1 + (1-2\eta)S^2][1 - (1-2\eta)S^2]}\Delta\mu(\nu). \tag{31}$$

*Proof of Theorem B.5.* By Lemma B.4, the conditional means of $\nu$ has the following forms.

$$\tilde{\mu}_0(\nu) := \mathbb{E}(\nu|\tilde{y} = 0)$$

$$= \int_\nu \nu \, d\mathrm{P}(\nu|\tilde{y} = 0)$$

$$= \int_\nu \nu \frac{(1 - \eta)(1 + S^2)}{(1 - \eta)(1 + S^2) + \eta(1 - S^2)} \, d\mathrm{P}(\nu|y = 0)$$

$$+ \int_\nu \nu \frac{\eta(1 - S^2)}{(1 - \eta)(1 + S^2) + \eta(1 - S^2)} \, d\mathrm{P}(\nu|y = 1). \tag{32}$$

$$\tilde{\mu}_1(\nu) := \mathbb{E}(\nu|\tilde{y} = 1)$$

$$= \int_\nu \nu \, d\mathrm{P}(\nu|\tilde{y} = 1)$$

$$= \int_\nu \nu \frac{\eta(1 + S^2)}{\eta(1 + S^2) + (1 - \eta)(1 - S^2)} \, d\mathrm{P}(\nu|y = 0)$$

$$+ \int_\nu \nu \frac{(1 - \eta)(1 - S^2)}{\eta(1 + S^2) + (1 - \eta)(1 - S^2)} \, d\mathrm{P}(\nu|y = 1). \tag{33}$$

Then we have

$$\tilde{\mu}_1(\nu) - \tilde{\mu}_0(\nu)$$

$$= \int_\nu \nu \left[ \frac{\eta(1 + S^2)}{\eta(1 + S^2) + (1 - \eta)(1 - S^2)} - \frac{(1 - \eta)(1 + S^2)}{(1 - \eta)(1 + S^2) + \eta(1 - S^2)} \right] d\mathrm{P}(\nu|y = 0)$$

$$+ \int_\nu \nu \left[ \frac{(1 - \eta)(1 - S^2)}{\eta(1 + S^2) + (1 - \eta)(1 - S^2)} - \frac{\eta(1 - S^2)}{(1 - \eta)(1 + S^2) + \eta(1 - S^2)} \right] d\mathrm{P}(\nu|y = 1)$$

$$= \frac{(1 - 2\eta)(1 + S^2)(1 - S^2)}{[1 + (1 - 2\eta)S^2][1 - (1 - 2\eta)S^2]} \left[ \int_\nu \nu \, d\mathrm{P}(\nu|y = 1) - \int_\nu \nu \, d\mathrm{P}(\nu|y = 0) \right]$$

$$= \frac{(1 - 2\eta)(1 + S^2)(1 - S^2)}{[1 + (1 - 2\eta)S^2][1 - (1 - 2\eta)S^2]} [\mu_1(\nu) - \mu_0(\nu)]. \tag{34}$$

$\square$

**Theorem B.6** (Influence of label noise on intra-class variance). *For $i = 0, 1$ and noise rate $\eta \in [0, \frac{1}{2})$,*

$$\tilde{\sigma}_i^2(\nu) = c_{i,0}\sigma_0^2(\nu) + c_{i,1}\sigma_1^2(\nu) + c_{i,0}\mu_0(\nu)^2 + c_{i,1}\mu_1(\nu)^2 - [c_{i,0}\mu_0(\nu) + c_{i,1}\mu_1(\nu)]^2$$

*where $c_{0,0} := \frac{(1-\eta)(1+S^2)}{1+(1-2\eta)S^2}$, $c_{0,1} := \frac{\eta(1+S^2)}{1+(1-2\eta)S^2}$, $c_{1,0} = \frac{\eta(1+S^2)}{1-(1-2\eta)S^2}$, and $c_{1,1} = \frac{(1-\eta)(1+S^2)}{1-(1-2\eta)S^2}$.*

*Proof of Theorem B.6.* By Lemma B.4, the conditional variances of $\nu$ has the following forms.

$$\tilde{\sigma}_0^2(\nu) := \mathbb{E}(\nu^2|\tilde{y} = 0) - \tilde{\mu}_0(\nu)^2$$

$$= \int_\nu \nu^2 \, d\mathrm{P}(\nu|\tilde{y} = 0) - \tilde{\mu}_0(\nu)^2$$

$$= \int_\nu \nu^2 \frac{(1 - \eta)(1 + S^2)}{(1 - \eta)(1 + S^2) + \eta(1 - S^2)} \, d\mathrm{P}(\nu|y = 0)$$

$$+ \int_\nu \nu^2 \frac{\eta(1 - S^2)}{(1 - \eta)(1 + S^2) + \eta(1 - S^2)} \, d\mathrm{P}(\nu|y = 1) - \tilde{\mu}_0(\nu)^2$$

$$= \frac{(1 - \eta)(1 + S^2)}{1 + (1 - 2\eta)S^2} [\sigma_0^2(\nu) + \mu_0(\nu)^2] + \frac{\eta(1 + S^2)}{1 + (1 - 2\eta)S^2} [\sigma_1^2(\nu) + \mu_1(\nu)^2]$$

$$- \left[ \frac{(1 - \eta)(1 + S^2)}{1 + (1 - 2\eta)S^2} \mu_0(\nu) + \frac{\eta(1 + S^2)}{1 + (1 - 2\eta)S^2} \mu_1(\nu) \right]^2$$

$$:= c_0\sigma_0^2(\nu) + c_1\sigma_1^2(\nu) + c_0\mu_0(\nu)^2 + c_1\mu_1(\nu)^2 - [c_0\mu_0(\nu) + c_1\mu_1(\nu)]^2, \tag{35}$$

where $c_0 := \frac{(1-\eta)(1+S^2)}{1+(1-2\eta)S^2}$ and $c_1 := \frac{\eta(1+S^2)}{1+(1-2\eta)S^2}$.

$$
\begin{aligned}
\tilde{\sigma}_1^2(\nu) &:= \mathbb{E}(\nu^2|\tilde{y}=1) - \tilde{\mu}_1(\nu)^2 \\
&= \int_\nu \nu^2 \, d\mathrm{P}(\nu|\tilde{y}=0) - \tilde{\mu}_0(\nu)^2 \\
&= \int_\nu \nu^2 \frac{\eta(1+S^2)}{\eta(1+S^2)+(1-\eta)(1-S^2)} \, d\mathrm{P}(\nu|y=0) \\
&\quad + \int_\nu \nu^2 \frac{(1-\eta)(1-S^2)}{\eta(1+S^2)+(1-\eta)(1-S^2)} \, d\mathrm{P}(\nu|y=1) - \tilde{\mu}_1(\nu)^2 \\
&= \frac{\eta(1+S^2)}{1-(1-2\eta)S^2}[\sigma_0^2(\nu)+\mu_0(\nu)^2] + \frac{(1-\eta)(1+S^2)}{1-(1-2\eta)S^2}[\sigma_1^2(\nu)+\mu_1(\nu)^2] \\
&\quad - \left[\frac{\eta(1+S^2)}{1-(1-2\eta)S^2}\mu_0(\nu) + \frac{(1-\eta)(1+S^2)}{1-(1-2\eta)S^2}\mu_1(\nu)\right]^2 \\
&:= c_0'\sigma_0^2(\nu) + c_1'\sigma_1^2(\nu) + c_0'\mu_0(\nu)^2 + c_1'\mu_1(\nu)^2 - [c_0'\mu_0(\nu)+c_1'\mu_1(\nu)]^2, \quad (36)
\end{aligned}
$$

where $c_0' = \frac{\eta(1+S^2)}{1-(1-2\eta)S^2}$ and $c_1' = \frac{(1-\eta)(1+S^2)}{1-(1-2\eta)S^2}$. □

### B.2.2 LINEAR SEPARABILITY OF MONOSEMANTIC & POLYSEMANTIC REPRESENTATIONS UNDER LABEL NOISE

*Proof of Theorem 4.2.* By definition, we have

$$
\frac{\tilde{J}(\nu_{\mathrm{mono}})/J(\nu_{\mathrm{mono}})}{\tilde{J}(\nu_{\mathrm{poly}})/J(\nu_{\mathrm{poly}})} = \frac{[\Delta\tilde{\mu}(\nu_{\mathrm{mono}})/(\tilde{\sigma}_0(\nu_{\mathrm{mono}})\tilde{\sigma}_1(\nu_{\mathrm{mono}}))]/[\Delta\mu(\nu_{\mathrm{mono}})/(\sigma_0(\nu_{\mathrm{mono}})\sigma_1(\nu_{\mathrm{mono}}))]}{[\Delta\tilde{\mu}(\nu_{\mathrm{poly}})/(\tilde{\sigma}_0(\nu_{\mathrm{poly}})\tilde{\sigma}_1(\nu_{\mathrm{poly}}))]/[\Delta\mu(\nu_{\mathrm{poly}})/(\sigma_0(\nu_{\mathrm{poly}})\sigma_1(\nu_{\mathrm{poly}}))]}.
\tag{37}
$$

By Theorem B.5 we have $\Delta\tilde{\mu}(\nu_{\mathrm{mono}})/\Delta\mu(\nu_{\mathrm{mono}}) = \Delta\tilde{\mu}(\nu_{\mathrm{poly}})/\Delta\mu(\nu_{\mathrm{poly}})$ and $\sigma_1(\nu_{\mathrm{mono}}) = \sigma_1(\nu_{\mathrm{poly}})$, and thus

$$
\frac{\tilde{J}(\nu_{\mathrm{mono}})/J(\nu_{\mathrm{mono}})}{\tilde{J}(\nu_{\mathrm{poly}})/J(\nu_{\mathrm{poly}})} = \frac{\tilde{\sigma}_0(\nu_{\mathrm{poly}})}{\tilde{\sigma}_0(\nu_{\mathrm{mono}})} \cdot \frac{\tilde{\sigma}_1(\nu_{\mathrm{poly}})}{\tilde{\sigma}_1(\nu_{\mathrm{mono}})} \cdot \frac{\sigma_0(\nu_{\mathrm{mono}})}{\sigma_0(\nu_{\mathrm{poly}})}.
\tag{38}
$$

By Theorems B.1 and B.6, we have

$$
\begin{aligned}
\tilde{\sigma}_0^2(\nu_{\mathrm{mono}}) &= \frac{1.04(1-\eta)}{1.04-0.08\eta}(0.246^2+0.205^2) + \frac{1.04\eta}{1.04-0.08\eta}(0.266^2+0.611^2) \\
&\quad - \left[\frac{1.04(1-\eta)}{1.04-0.08\eta}0.205 + \frac{1.04\eta}{1.04-0.08\eta}0.611\right]^2,
\end{aligned}
\tag{39}
$$

$$
\begin{aligned}
\tilde{\sigma}_1^2(\nu_{\mathrm{mono}}) &= \frac{1.04\eta}{0.96+0.08\eta}(0.246^2+0.205^2) + \frac{1.04(1-\eta)}{0.96+0.08\eta}(0.266^2+0.611^2) \\
&\quad - \left[\frac{1.04\eta}{0.96+0.08\eta}0.205 + \frac{1.04(1-\eta)}{0.96+0.08\eta}0.611\right]^2,
\end{aligned}
\tag{40}
$$

$$
\begin{aligned}
\tilde{\sigma}_0^2(\nu_{\mathrm{poly}}) &= \frac{1.04(1-\eta)}{1.04-0.08\eta}(0.276^2+(-0.359)^2) + \frac{1.04\eta}{1.04-0.08\eta}(0.266^2+0.389^2) \\
&\quad - \left[\frac{1.04(1-\eta)}{1.04-0.08\eta}(-0.359)) + \frac{1.04\eta}{1.04-0.08\eta}0.389\right]^2,
\end{aligned}
\tag{41}
$$

$$
\begin{aligned}
\tilde{\sigma}_1^2(\nu_{\mathrm{poly}}) &= \frac{1.04\eta}{0.96+0.08\eta}(0.276^2+(-0.359)^2) + \frac{1.04(1-\eta)}{0.96+0.08\eta}(0.266^2+0.389^2) \\
&\quad - \left[\frac{1.04\eta}{0.96+0.08\eta}(-0.359) + \frac{1.04(1-\eta)}{0.96+0.08\eta}0.389\right]^2.
\end{aligned}
\tag{42}
$$

Then plugging Eq. (39), Eq. (40), Eq. (41), and Eq. (42) into Eq. (38), we have $\frac{\tilde{J}(\nu_{\mathrm{mono}})/J(\nu_{\mathrm{mono}})}{\tilde{J}(\nu_{\mathrm{poly}})/J(\nu_{\mathrm{poly}})} \geq 1$.

Further, by Theorems B.1 and B.5, we have

$$\Delta\tilde{\mu}(\nu_{\mathrm{mono}}) = \frac{1.04 \times 0.96(1 - 2\eta)}{(1.04 - 0.08\eta)(0.96 + 0.08\eta)} \times 0.406, \tag{43}$$

$$\Delta\tilde{\mu}(\nu_{\mathrm{poly}}) = \frac{1.04 \times 0.96(1 - 2\eta)}{(1.04 - 0.08\eta)(0.96 + 0.08\eta)} \times 0.748. \tag{44}$$

Plugging them into the definition of $\tilde{J}(\nu_{\mathrm{mono}})$ and $\tilde{J}(\nu_{\mathrm{poly}})$, we have $\tilde{J}(\nu_{\mathrm{mono}}) < \tilde{J}(\nu_{\mathrm{poly}})$ when $\eta < 0.25$ and $\tilde{J}(\nu_{\mathrm{mono}}) > \tilde{J}(\nu_{\mathrm{poly}})$ when $\eta > 0.25$.

$\square$

### B.3 PROOFS RELATED TO INPUT NOISE

Following the settings in Section 4.2, we investigate the influence of Gaussian noise $\varepsilon_i \sim \mathcal{N}(0, 1)$, i.i.d. $i = 1, 2$, on the input data $x = (x_1, x_2)$, where $\varepsilon_i \perp x$. Given noise strength $\lambda > 0$, we denote the noisy input data as $\tilde{x} = (x_1 + \lambda\varepsilon_1, x_2 + \lambda\varepsilon_2)$. Then the learned monosemantic and polysemantic representations are $\nu_{\mathrm{mono}} = x_1 + \lambda\varepsilon_1$ and $\nu_{\mathrm{poly}} = (x_1 - x_2) + \lambda(\varepsilon_1 - \varepsilon_2)$. Next, we derive the influence of noise strength on the conditional means and variances, respectively.

**Theorem B.7** (Influence of Gaussian noise on inter-class distance). *Given noise strength $\lambda > 0$, for both mono- and poly-semantic representations, we have*

$$\Delta\tilde{\mu}(\nu) = \Delta\mu(\nu). \tag{45}$$

*Proof of Theorem B.7.* For $\nu_{\mathrm{mono}}$ and $i = 0, 1$,

$$\begin{aligned}
\tilde{\mu}_i(\nu_{\mathrm{mono}}) &= \mathbb{E}(x_1 + \lambda\varepsilon_1 | y = i) \\
&= \mathbb{E}(x_1 | y = i) + \lambda\mathbb{E}(\varepsilon_1 | y = i) \\
&= \mathbb{E}(x_1 | y = i) + 0 \\
&= \mu_i(\nu_{\mathrm{mono}}).
\end{aligned} \tag{46}$$

For $\nu_{\mathrm{poly}}$ and $i = 0, 1$,

$$\begin{aligned}
\tilde{\mu}_i(\nu_{\mathrm{poly}}) &= \mathbb{E}((x_1 - x_2) + \lambda(\varepsilon_1 - \varepsilon_2) | y = i) \\
&= \mathbb{E}(x_1 - x_2 | y = i) + \lambda[\mathbb{E}(\varepsilon_1 | y = i) - \mathbb{E}(\varepsilon_2 | y = i)] \\
&= \mathbb{E}(x_1 - x_2 | y = i) + 0 \\
&= \mu_i(\nu_{\mathrm{poly}}).
\end{aligned} \tag{47}$$

Then for $\nu \in \{\nu_{\mathrm{mono}}, \nu_{\mathrm{poly}}\}$,

$$\Delta\tilde{\mu}(\nu) = \tilde{\mu}_1(\nu) - \tilde{\mu}_0(\nu) = \mu_1(\nu) - \mu_0(\nu) = \Delta\mu(\nu). \tag{48}$$

$\square$

**Theorem B.8** (Influence of Gaussian noise on intra-class variance). *For $i = 0, 1$ and noise strength $\lambda > 0$, we have*

$$\tilde{\sigma}_i^2(\nu_{\mathrm{mono}}) = \sigma_i^2(\nu_{\mathrm{mono}}) + \lambda^2, \tag{49}$$

*and*

$$\tilde{\sigma}_i^2(\nu_{\mathrm{poly}}) = \sigma_i^2(\nu_{\mathrm{poly}}) + 2\lambda^2. \tag{50}$$

*Proof of Theorem B.8.* For $\nu_{\mathrm{mono}}$ and $i = 0, 1$,

$$\begin{aligned}
\tilde{\sigma}_i^2(\nu_{\mathrm{mono}}) &= \mathbb{E}((x_1 + \lambda\varepsilon_1)^2 | y = i) - \tilde{\mu}_i(\nu_{\mathrm{mono}}) \\
&= \mathbb{E}(x_1^2 | y = i) + 2\lambda\mathbb{E}(x_1\varepsilon_1 | y = i) + \lambda^2\mathbb{E}(\varepsilon_1^2 | y = i) - \tilde{\mu}_i(\nu_{\mathrm{mono}}) \\
&= \mathbb{E}(x_1^2 | y = i) - \tilde{\mu}_i(\nu_{\mathrm{mono}}) + 0 + \lambda^2\mathbb{E}(\varepsilon_1^2 | y = i) \\
&= \sigma_i^2(\nu_{\mathrm{mono}}) + \lambda^2.
\end{aligned} \tag{51}$$

For $\nu_{\mathrm{poly}}$ and $i = 0, 1$,

$$
\begin{aligned}
\tilde{\sigma}_i^2(\nu_{\mathrm{mono}}) &= \mathbb{E}(((x_1 - x_2) + \lambda(\varepsilon_1 - \varepsilon_2))^2 | y = i) - \tilde{\mu}_i(\nu_{\mathrm{poly}}) \\
&= \mathbb{E}((x_1 - x_2)^2 | y = i) + 2\lambda\mathbb{E}((x_1 - x_2)(\varepsilon_1 - \varepsilon_2) | y = i) + \lambda^2\mathbb{E}((\varepsilon_1 - \varepsilon_2)^2 | y = i) - \tilde{\mu}_i(\nu_{\mathrm{poly}}) \\
&= \mathbb{E}((x_1 - x_2)^2 | y = i) - \tilde{\mu}_i(\nu_{\mathrm{poly}}) + 2\lambda\mathbb{E}((x_1 - x_2)\varepsilon_1 | y = i) - 2\lambda\mathbb{E}((x_1 - x_2)\varepsilon_2 | y = i) \\
&\quad + \lambda^2\mathbb{E}(\varepsilon_1^2 | y = i) - 2\lambda^2\mathbb{E}(\varepsilon_1\varepsilon_2 | y = i) + \lambda^2\mathbb{E}(\varepsilon_2^2 | y = i) \\
&= \sigma_i^2(\nu_{\mathrm{poly}}) + 2\lambda^2.
\end{aligned} \tag{52}
$$

$\square$

**Theorem B.9** (Influence of Gaussian noise on linear seprarability). *We denote the linear separability criterion under noise as $\tilde{J}(\nu) = \Delta\tilde{\mu}(\nu)/(\tilde{\sigma}_0(\nu)\tilde{\sigma}_1(\nu))$. For noise rate $\lambda > 0$,*

$$
\frac{\tilde{J}(\nu_{\mathrm{poly}})}{J(\nu_{\mathrm{poly}})} \leq \frac{\tilde{J}(\nu_{\mathrm{mono}})}{J(\nu_{\mathrm{mono}})} \leq 1. \tag{53}
$$

*Meanwhile, we obtain $\tilde{J}(\nu_{\mathrm{poly}}) \leq \tilde{J}(\nu_{\mathrm{mono}})$ when $\lambda \geq 0.55$.*

As shown in Theorem B.9, with the increase of noise strength, the linear separability ($J(\nu)$) of both polysemantic and monosemantic features becomes worse. However, $J(\nu_{mono})$ decreases more slowly. As a result, when the noise strength is aggressive enough ($\lambda \geq 0.25$), the monosemantic feature exhibits better linear seperability than the polysemantic one. The theoretical results reveal that the linear separability of monosemantic features is more robust than polysemantic features, which leads to better performance in tasks under Input noise.

*Proof of Theorem B.9.* By definition, we have

$$
\frac{\tilde{J}(\nu_{\mathrm{mono}})/J(\nu_{\mathrm{mono}})}{\tilde{J}(\nu_{\mathrm{poly}})/J(\nu_{\mathrm{poly}})} = \frac{[\Delta\tilde{\mu}(\nu_{\mathrm{mono}})/(\tilde{\sigma}_0(\nu_{\mathrm{mono}})\tilde{\sigma}_1(\nu_{\mathrm{mono}}))]/[\Delta\mu(\nu_{\mathrm{mono}})/(\sigma_0(\nu_{\mathrm{mono}})\sigma_1(\nu_{\mathrm{mono}}))]}{[\Delta\tilde{\mu}(\nu_{\mathrm{poly}})/(\tilde{\sigma}_0(\nu_{\mathrm{poly}})\tilde{\sigma}_1(\nu_{\mathrm{poly}}))]/[\Delta\mu(\nu_{\mathrm{poly}})/(\sigma_0(\nu_{\mathrm{poly}})\sigma_1(\nu_{\mathrm{poly}}))]}. \tag{54}
$$

By Theorems B.7 and B.8, we have $\Delta\tilde{\mu}(\nu_{\mathrm{mono}}) = \Delta\mu(\nu_{\mathrm{mono}})$, $\Delta\tilde{\mu}(\nu_{\mathrm{poly}}) = \Delta\mu(\nu_{\mathrm{poly}})$, $\tilde{\sigma}_i^2(\nu_{\mathrm{mono}}) = \sigma_i^2(\nu_{\mathrm{mono}}) + \lambda^2$, and $\tilde{\sigma}_i^2(\nu_{\mathrm{poly}}) = \sigma_i^2(\nu_{\mathrm{poly}}) + 2\lambda^2$, $i = 0, 1$. By Theorem 4.1, we have $\sigma_1(\nu_{\mathrm{mono}}) = \sigma_1(\nu_{\mathrm{poly}})$. Then we have

$$
\begin{aligned}
\frac{\tilde{J}(\nu_{\mathrm{mono}})/J(\nu_{\mathrm{mono}})}{\tilde{J}(\nu_{\mathrm{poly}})/J(\nu_{\mathrm{poly}})} &= \frac{\tilde{\sigma}_0(\nu_{\mathrm{poly}})\tilde{\sigma}_1(\nu_{\mathrm{poly}})\sigma_0(\nu_{\mathrm{mono}})}{\tilde{\sigma}_0(\nu_{\mathrm{mono}})\tilde{\sigma}_1(\nu_{\mathrm{mono}})\sigma_0(\nu_{\mathrm{poly}})} \\
&= \frac{\sqrt{(\sigma_0^2(\nu_{\mathrm{poly}}) + 2\lambda^2)(\sigma_1^2(\nu_{\mathrm{poly}}) + 2\lambda^2)}\sigma_0(\nu_{\mathrm{mono}})}{\sqrt{(\sigma_0^2(\nu_{\mathrm{mono}}) + \lambda^2)(\sigma_1^2(\nu_{\mathrm{mono}}) + \lambda^2)}\sigma_0(\nu_{\mathrm{poly}})}.
\end{aligned} \tag{55}
$$

Then plugging Theorem 4.1, we complete the proof. $\square$

### B.4 LINEAR SEPARABILITY OF POLYSEMANTIC REPRESENTATIONS (GENERALIZED FORM)

In this part, we generalize the polysemantic representation to $\nu_{\mathrm{poly}} = w_1 x_1 - w_2 x_2$, where $w_1, w_2 > 0$. Without loss of generality, we could assume $w_1 \geq w_2$. The case $w_1 < w_2$ is equivalent to study $\nu'_{\mathrm{poly}} = -\nu_{\mathrm{poly}} = w_2 x_2 - w_1 x_1$, whose distribution is the same as $w_2 x_1 - w_1 x_2$ because $x_1$ and $x_2$ are i.i.d. distributed. Note that when $w_1 = 1$, $w_2 = 0$, $\nu_{\mathrm{poly}} = \nu_{\mathrm{mono}}$, and when $w_1 = w_2 = 1$, the results in this section reduce to Theorems 4.1 and 4.2.

We first calculate some related probability distribution functions in Lemma B.10.

**Lemma B.10.**

$$P(x_1 \le \min\{x/(w_1 - w_2), x_2\})$$
$$= \begin{cases} 0, & \text{if } x \in [-w_2, 0), \\ -\dfrac{1}{2}(1-S)^2(1-x/(w_1-w_2))^2 + \dfrac{1}{2}(1+S^2), & \text{if } x \in [0, w_1 - w_2], \\ \dfrac{1}{2}(1+S^2), & \text{if } x \in (w_1 - w_2, w_1] \end{cases} \quad (56)$$

$$P(x/(w_1 - w_2) < x_1 \le (x + w_2 x_2)/w_1)$$
$$= \begin{cases} \dfrac{w_2}{2w_1}[(1-S)(1+x/w_2) + Sw_1/w_2]^2 - \dfrac{w_1}{2w_2}S^2, & \text{if } x \in [-w_2, 0), \\ \dfrac{w_2}{2w_1}(1-S)^2[1-x/(w_1-w_2)]^2, & \text{if } x \in [0, w_1 - w_2], \\ 0, & \text{if } x \in (w_1 - w_2, w_1]. \end{cases} \quad (57)$$

$$P(x_2 < x_1 \le \min\{(x + w_2 x_2)/w_1, x/(w_1 - w_2)\})$$
$$= \begin{cases} 0, & \text{if } x \in [-w_2, 0), \\ \dfrac{1}{2w_1(w_1-w_2)}[(1-S)x + S(w_1-w_2)]^2 - \dfrac{w_1 - w_2}{2w_1}S^2, & \text{if } x \in [0, w_1 - w_2], \\ -\dfrac{w_1}{2w_2}(1-S)^2(1-x/w_1)^2 + S(1-S)x/w_1 + \dfrac{1}{2}(1-S)^2, & \text{if } x \in (w_1 - w_2, w_1]. \end{cases} \quad (58)$$

Then we can calculate the conditional mean and variance of $\nu_{\text{poly}} = w_1 x_1 - w_2 x_2$.

**Theorem B.11** (Conditional mean and variance of polysemantic representations)**.** *The conditional means and variances of $\nu_{\text{poly}} = w_1 x_1 - w_2 x_2$ are*

$$\mu_0(\nu_{\text{poly}}) = \frac{1-S}{3(1+S^2)}[(w_1 - 2w_2) - (w_1 + w_2)S] \quad (59)$$

$$\mu_1(\nu_{\text{mono}}) = \frac{(2w_1 - w_2) + (w_1 + w_2)S}{3(1+S)} \quad (60)$$

$$\sigma_0^2(\nu_{\text{poly}}) = \frac{1-S}{6(1+S^2)}[(w_1^2 - 3w_1 w_2 + 3w_2^2) - (w_1^2 - 3w_1 w_2 - w_2^2)S] - \mu_0(\nu_{\text{poly}})^2, \quad (61)$$

$$\sigma_1^2(\nu_{\text{poly}}) = \frac{(3w_1^2 - 3w_1 w_2 + w_2^2) + (w_1^2 + 3w_1 w_2 - w_2^2)S}{6(1+S)} - \mu_1(\nu_{\text{poly}})^2. \quad (62)$$

*Proof of Theorem B.3.* If $w_1 \ge w_2$, by Lemma B.10, we have

$$P_{\text{poly}}(w_1 x_1 - w_2 x_2 \le x | y = 0)$$
$$= P(w_1 x_1 - w_2 x_2 \le x | x_1 \le x_2)$$
$$= \frac{P(x_1 \le \min\{x/(w_1 - w_2), x_2\}) + P(x/(w_1 - w_2) < x_1 \le (x + w_2 x_2)/w_1)}{P(x_1 - x_2 \le 0)}$$
$$= \begin{cases} \dfrac{w_2}{w_1}\dfrac{1}{1+S^2}[(1-S)(1+x/w_2) + Sw_1/w_2]^2 - \dfrac{w_1}{w_2}\dfrac{S^2}{1+S^2}, & \text{if } x \in [-w_2, 0), \\ -(1 - w_2/w_1)\dfrac{(1-S)^2}{1+S^2}(1-x/(w_1-w_2))^2 + 1, & \text{if } x \in [0, w_1 - w_2], \\ 1, & \text{if } x \in (w_1 - w_2, w_1] \end{cases} \quad (63)$$

and

$$P_{poly}(w_1 x_1 - w_2 x_2 \le x | y = 1)$$
$$= P(w_1 x_1 - w_2 x_2 \le x | x_1 > x_2)$$
$$= \frac{P(x_2 < x_1 \le \min\{(x + w_2 x_2)/w_1, x/(w_1 - w_2)\})}{P(x_1 - x_2 > 0)}$$
$$= \begin{cases} 0, & \text{if } x \in [-w_2, 0), \\ \frac{1}{w_1(w_1 - w_2)} \frac{1}{1 - S^2} [(1 - S)x + S(w_1 - w_2)]^2 - \frac{w_1 - w_2}{w_1} \frac{S^2}{1 - S^2}, & \text{if } x \in [0, w_1 - w_2], \\ -\frac{w_1}{w_2} \frac{1 - S}{1 + S} (1 - x/w_1)^2 + \frac{2S}{1 + S} x/w_1 + \frac{(1 - S)}{(1 + S)}, & \text{if } x \in (w_1 - w_2, w_1]. \end{cases}$$

$$(64)$$

Then we have

$$\mu_0(\nu_{poly}) = \int_{x \in [-w_2, 0)} x \cdot \frac{2}{w_1} \frac{1 - S}{1 + S^2} [(1 - S)(1 + x/w_2) + S w_1/w_2] \, dx$$
$$+ \int_{x \in [0, w_1 - w_2]} x \cdot \frac{2}{w_1} \frac{(1 - S)^2}{1 + S^2} [1 - x/(w_1 - w_2)] \, dx$$
$$= -\frac{3S(1 - S)w_1 w_2 + (1 - S)^2 w_2^2}{3(1 + S^2)w_1} + \frac{(1 - S)^2 (w_1 - w_2)^2}{3(1 + S^2)w_1}$$
$$= \frac{1 - S}{3(1 + S^2)} [(1 - S)w_1 - (2 + S)w_2]$$
$$= \frac{1 - S}{3(1 + S^2)} [(w_1 - 2w_2) - (w_1 + w_2)S],$$

$$(65)$$

$$\mu_1(\nu_{poly}) = \int_{x \in [0, w_1 - w_2]} x \cdot \frac{2}{w_1(w_1 - w_2)(1 + S)} [(1 - S)x + (w_1 - w_2)S] \, dx$$
$$+ \int_{x \in (w_1 - w_2, w_1]} x \cdot \left[ \frac{2}{w_2} \frac{1 - S}{1 + S} (1 - x/w_1) + \frac{2}{w_1} \frac{S}{1 + S} \right] \, dx$$
$$= \frac{(2 + S)(w_1 - w_2)^2}{3(1 + S)w_1} + \frac{3(1 + S)w_1 w_2 - (2 + S)w_2^2}{3(1 + S)w_1}$$
$$= \frac{(2 + S)w_1 - (1 - S)w_2}{3(1 + S)}$$
$$= \frac{(2w_1 - w_2) + (w_1 + w_2)S}{3(1 + S)},$$

$$(66)$$

$$\mu_1(\nu_{poly}) - \mu_0(\nu_{poly}) = \frac{(w_1 + w_2) + 2(w_1 + w_2)S + 3(w_1 - w_2)S^2}{3(1 + S)(1 + S^2)}, \qquad (67)$$

$$\sigma_0^2(\nu_{\text{poly}}) = \int_{x\in[-w_2,0)} x^2 \cdot \frac{2}{w_1}\frac{1-S}{1+S^2}[(1-S)(1+x/w_2)+Sw_1/w_2]\,dx$$

$$+ \int_{x\in[0,w_1-w_2]} x^2 \cdot \frac{2}{w_1}\frac{(1-S)^2}{1+S^2}[1-x/(w_1-w_2)]\,dx - \mu_0(\nu_{\text{poly}})^2$$

$$= \frac{4S(1-S)w_1w_2^2+(1-S)^2w_2^3}{6(1+S^2)w_1}$$

$$+ \frac{(1-S)^2w_1^3-3(1-S)^2w_1^2w_2+3(1-S)^2w_1w_2^2-(1-S)^2w_2^3}{6(1+S^2)w_1} - \mu_0(\nu_{\text{poly}})^2$$

$$= \frac{(1-S)^2w_1^2-3(1-S)^2w_1w_2+(1-S)(3+S)w_2^2}{6(1+S^2)} - \mu_0(\nu_{\text{poly}})^2$$

$$= \frac{1-S}{6(1+S^2)}[(1-S)w_1^2-3(1-S)w_1w_2+(3+S)w_2^2] - \mu_0(\nu_{\text{poly}})^2$$

$$= \frac{1-S}{6(1+S^2)}[(w_1^2-3w_1w_2+3w_2^2)-(w_1^2-3w_1w_2-w_2^2)S] - \mu_0(\nu_{\text{poly}})^2, \quad (68)$$

and

$$\sigma_1^2(\nu_{\text{poly}}) = \int_{x\in[0,w_1-w_2]} x^2 \cdot \frac{2}{w_1(w_1-w_2)(1+S)}[(1-S)x+(w_1-w_2)S]\,dx$$

$$+ \int_{x\in(w_1-w_2,w_1]} x^2 \cdot \left[\frac{2}{w_2}\frac{1-S}{1+S}(1-x/w_1)+\frac{2}{w_1}\frac{S}{1+S}\right]dx - \mu_1(\nu_{\text{poly}})^2$$

$$= \frac{(3+S)w_1^3-3(3+S)w_1^2w_2+3(3+S)w_1w_2^2-(3+S)w_2^3}{6(1+S)w_1}$$

$$+ \frac{6(1+S)w_1^2w_2-4(2+S)w_1w_2^2+(3+S)w_2^3}{6(1+S)w_1} - \mu_1(\nu_{\text{poly}})^2$$

$$= \frac{(3+S)w_1^2-3(1-S)w_1w_2+(1-S)w_2^2}{6(1+S)} - \mu_1(\nu_{\text{poly}})^2$$

$$= \frac{(3w_1^2-3w_1w_2+w_2^2)+(w_1^2+3w_1w_2-w_2^2)S}{6(1+S)} - \mu_1(\nu_{\text{poly}})^2. \quad (69)$$

$\square$

Following the toy model described in Section 4.2, where we let $S = 0.2$, we compare the linear separability of monosemantic and polysemantic features in Theorem B.12.

**Theorem B.12** (Conditional means and variances of monosemantic & polysemantic features (Generalized Form)). *If $0 < w_2 \le w_1 = 1$, for conditional means, we have $\mu_0(\nu_{\text{poly}}) < \mu_0(\nu_{\text{mono}})$ and $\mu_1(\nu_{\text{poly}}) < \mu_1(\nu_{\text{mono}})$, yet $\Delta\mu(\nu_{\text{poly}}) > \Delta\mu(\nu_{\text{mono}})$. For conditional variances, we have $\sigma_1^2(\nu_{\text{poly}}) \le \sigma_1^2(\nu_{\text{mono}})$ and $\sigma_0^2(\nu_{\text{poly}}) > \sigma_0^2(\nu_{\text{mono}})$. Overall, we have $J(\nu_{\text{poly}}) > J(\nu_{\text{mono}})$.*

*Proof of Theorem B.12.* By Theorem B.1, we have $\mu_0(\nu_{\text{mono}}) = 0.205$, $\mu_1(\nu_{\text{mono}}) = 0.611$, $\Delta\mu(\nu_{\text{mono}}) = 0.406$, $\sigma_0^2(\nu_{\text{mono}}) = 0.061$, $\sigma_1^2(\nu_{\text{mono}}) = 0.071$, and $J(\nu_{\text{mono}}) = 6.196$. By Theorem B.11, we have $\mu_0(\nu_{\text{poly}}) = 0.205w_1-0.564w_2$, $\mu_1(\nu_{\text{poly}}) = 0.611w_1-0.222w_2$, $\Delta\mu(\nu_{\text{poly}}) = 0.406w_1 + 0.342w_2$, $\sigma_0^2(\nu_{\text{poly}}) = 0.061w_1^2 - 0.077w_1w_2 + 0.092w_2^2$, $\sigma_1^2(\nu_{\text{poly}}) = 0.071w_1^2 - 0.062w_1w_2+0.062w_2^2$, and $J(\nu_{\text{poly}}) = \frac{0.406w_1+0.342w_2}{\sqrt{(0.061w_1^2-0.077w_1w_2+0.092w_2^2)(0.071w_1^2-0.062w_1w_2+0.062w_2^2)}}$. By comparing the above results, we complete the proof. $\square$

**Theorem B.13** (Influence of label noise on linear seprarability (Generalized Form)). *We denote the linear separability criterion under noise as $\tilde{J}(\nu) = \Delta\tilde{\mu}(\nu)/(\tilde{\sigma}_0(\nu)\tilde{\sigma}_1(\nu))$. If $w_2 \le w_1 = 1$, then for noise rate $\eta \in [0, 0.5)$, we have*

$$\frac{\tilde{J}(\nu_{\text{poly}})}{J(\nu_{\text{poly}})} \le \frac{\tilde{J}(\nu_{\text{mono}})}{J(\nu_{\text{mono}})} \le 1. \quad (70)$$

*Proof of Theorem B.13.* By definition, we have

$$\frac{\tilde{J}(\nu_{\text{mono}})/J(\nu_{\text{mono}})}{\tilde{J}(\nu_{\text{poly}})/J(\nu_{\text{poly}})} = \frac{[\Delta\tilde{\mu}(\nu_{\text{mono}})/(\tilde{\sigma}_0(\nu_{\text{mono}})\tilde{\sigma}_1(\nu_{\text{mono}}))]/[\Delta\mu(\nu_{\text{mono}})/(\sigma_0(\nu_{\text{mono}})\sigma_1(\nu_{\text{mono}}))]}{[\Delta\tilde{\mu}(\nu_{\text{poly}})/(\tilde{\sigma}_0(\nu_{\text{poly}})\tilde{\sigma}_1(\nu_{\text{poly}}))]/[\Delta\mu(\nu_{\text{poly}})/(\sigma_0(\nu_{\text{poly}})\sigma_1(\nu_{\text{poly}}))]}.$$
(71)

By Theorem B.5 we have $\Delta\tilde{\mu}(\nu_{\text{mono}})/\Delta\mu(\nu_{\text{mono}}) = \Delta\tilde{\mu}(\nu_{\text{poly}})/\Delta\mu(\nu_{\text{poly}})$, and thus

$$\frac{\tilde{J}(\nu_{\text{mono}})/J(\nu_{\text{mono}})}{\tilde{J}(\nu_{\text{poly}})/J(\nu_{\text{poly}})} = \frac{\tilde{\sigma}_0(\nu_{\text{poly}})}{\tilde{\sigma}_0(\nu_{\text{mono}})} \cdot \frac{\tilde{\sigma}_1(\nu_{\text{poly}})}{\tilde{\sigma}_1(\nu_{\text{mono}})} \cdot \frac{\sigma_0(\nu_{\text{mono}})}{\sigma_0(\nu_{\text{poly}})} \cdot \frac{\sigma_1(\nu_{\text{mono}})}{\sigma_1(\nu_{\text{poly}})}.$$
(72)

Then given $\eta \in [0, 0.5)$, by Theorems B.1, B.6, B.11, we can calculate the specific values of $\tilde{\sigma}_0(\nu_{\text{mono}})$, $\tilde{\sigma}_1(\nu_{\text{mono}})$, $\tilde{\sigma}_0(\nu_{\text{poly}})$, $\tilde{\sigma}_1(\nu_{\text{poly}})$, and correspondingly we have $\frac{\tilde{J}(\nu_{\text{mono}})/J(\nu_{\text{mono}})}{\tilde{J}(\nu_{\text{poly}})/J(\nu_{\text{poly}})} \geq 1$. □

### B.5 MONOSEMANTICITY OF REPRESENTATIONS LEARNED BY NON-NEGATIVE CONTRASTIVE LEARNING AND NON-NEGATIVE TUNING

Wang et al. (2024) proved that the optimal solutions of non-negative contrastive learning are monosemantic representations. We first introduce some necessary notations. Following (Saunshi et al., 2019), they assume that samples in the pretraining data belong to $m$ latent classes $\mathcal{C} = \{c_1, \cdots, c_m\}$. Besides that, they assume that positive samples in contrastive learning are darw independently from the same latent class, i.e.,

**Assumption B.14** (Positive Generation). $\forall x, x' \in \mathcal{X}, \mathcal{P}(x, x') = \mathbb{E}_c \mathcal{P}(x|c)\mathcal{P}(x'|c).$

Then they propose the following Lemma that introduces the optimal solutions of non-negative contrastive learning.

**Lemma B.15** ((Wang et al., 2024)). *We denote that*

$$\phi(x) = \left[\frac{1}{\sqrt{\mathcal{P}(\pi_1)}}\mathcal{P}(\pi_1|x), \ldots, \frac{1}{\sqrt{\mathcal{P}(\pi_m)}}\mathcal{P}(\pi_m|x)\right] \in \mathbb{R}_+^m, \forall\, x \in \mathcal{X},$$
(73)

*where $[\pi_1, \cdots, \pi_m]$ is a random permutation of latent classes $[c_1, \cdots, c_m]]$. Under the latent class assumption and choosing $k = m$, $\phi(\cdot)$ is a minimizer of the NCL objective, i.e., $\phi \in \arg\min \mathcal{L}_{\text{NCL}}$.*

As shown in the lemma above, the samples activated in the same dimension belong to the same latent classes, which means the optimal solutions are monosemantic. In the next step, we extend the theory to non-negative tuning. Following (Wang et al., 2024), we use the spectral loss for the simplicity of analysis, i.e. $L_{\text{SCE}} = -\mathbb{E}_{(x,y(x))}(Wf(x))^\top \mathbb{1}_{y(x)} + 2\mathbb{E}_{x,y_i}((Wf(x))^\top \mathbb{1}_{y_i})^2$, where $f(x)$ is the representation layer, $W$ is the linear classifier, $y(x)$ is the label of $x$ and there exist $t$ classes. Respectively, with the non-negative constraints, the objective becomes $L_{\text{SNCE}} = -\mathbb{E}_{(x,y(x))}(Wf_+(x))^\top \mathbb{1}_{y(x)} + 2\mathbb{E}_{x,y_i}((Wf_+(x))^\top \mathbb{1}_{y_i})^2$. When each sample $x$ has a ground-truth label $y(x)$, we obtain the following theorem:

**Theorem B.16.** *We denote that*

$$\beta(x) = \left[\frac{1}{\sqrt{\mathcal{P}(y_1)}}\mathcal{P}(y_1|x), \ldots, \frac{1}{\sqrt{\mathcal{P}(y_t)}}\mathcal{P}(y_t|x)\right] \in \mathbb{R}_+^t, \forall\, x \in \mathcal{X},$$
(74)

*When we choose $t = k$, $\beta(\cdot)$ is a minimizer of the non-negative tuning objective, i.e., $\phi \in \arg\min \mathcal{L}_{\text{SNCE}}$.*

Consequently, we find that the optimal solutions of non-negative tuning are also monosemantic as each dimension is only activated by the samples in the same class.

*Proof.* We first prove that the non-negative tuning objective is equal to a matrix decomposition objective. To be specific, Let $\bar{A}$ be the normalized co-occurrence matrix, i.e., $\bar{A}_{x,y(x)} = \frac{A_{x,y(x)}}{\sqrt{P(x)P(y(x))}}$, where $P(x), P(y(x))$ denote the marginal distribution.

Then we denote a matrix decomposition objective as

$$\mathcal{L}_{MF} = \|\bar{A} - FW'^{\top}\|^2 + const.$$

where the $x$-th row of $F$ and the $y(x)$-th row of $W'$ respectively represents representation features and the linear classifier, i.e., $F_x = \sqrt{C(x)}f_+(x)^{\top}$, $W'_{y(x)} = \sqrt{P(y(x))}W_{y(x)}$. Then we expand the objective:

$$
\begin{aligned}
\|\bar{A} - FW'^{\top}\|^2 &= \sum_{x,y(x)} \left(\bar{A}_{x,y(x)} - F_x(W'_{y(x)})^{\top}\right)^2 \\
&= \sum_{x,y(x)} \left(\frac{P(x,y(x))}{\sqrt{P(x)P(y(x))}} - \sqrt{P(x)}f_+(x)^{\top}\sqrt{P(y(x))}(W_{y(x)})^{\top}\right)^2 \\
&= \sum_{x,y(x)} \left(\frac{P(x,y(x))^2}{P(x)P(y(x))} + P(x)P(y(x))\left(f_+(x)^{\top}(W_{y(x)})^{\top}\right)^2 - 2P(x,y(x))f_+(x)^{\top}(W_{y(x)})^{\top}\right) \\
&= \sum_{x,y(x)} \left(\frac{P(x,y(x))^2}{P(x)P(y(x))}\right) - 2\mathbb{E}_{(x,y(x))}f_+(x)^{\top}(W_{y(x)})^{\top} + \mathbb{E}_{x,y_i}\left(f_+(x)^{\top}(W_{y_i})^{\top}\right)^2 \\
&= \sum_{x,y(x)} \left(\frac{P(x,y(x))^2}{P(x)P(y(x))}\right) - 2\mathbb{E}_{(x,y(x))}(Wf_+(x))^{\top}\mathbb{1}_{y(x)} + \mathbb{E}_{x,y_i}((Wf_+(x))^{\top}\mathbb{1}_{y_i})^2 \\
&= \mathcal{L}_{\text{SNCE}} + const.
\end{aligned}
$$

Then we expand $L_{MF}(\beta)$:

$$\|\bar{A} - FW'^{\top}\|^2 = \sum_{x,y(x)} \left(\frac{P(x,y(x))}{\sqrt{P(x)P(y(x))}} - \sqrt{P(x)}\beta(x)^{\top}\sqrt{P(y(x))}(W_{y(x)})^{\top}\right)^2$$

When $W$ satisfies $W_{y_i} = \frac{1}{\sqrt{P(y_i)}}$, we obtain $\qquad\qquad\qquad\qquad\qquad\qquad\qquad\square$

$$
\begin{aligned}
\|\bar{A} - FW'^{\top}\|^2 &= \sum_{x,y(x)} \left(\frac{P(x,y(x))}{\sqrt{P(x)P(y(x))}} - \sqrt{P(x)}\beta(x)^{\top}\right)^2 \\
&= \sum_{x,y(x)} \left(\frac{P(x,y(x))}{\sqrt{P(x)P(y(x))}} - \frac{\sqrt{P(x)}p(y(x)|x)}{\sqrt{P(y(x))}}\right)^2 \\
&= \sum_{x,y(x)} \left(\frac{P(x,y(x))}{\sqrt{P(x)P(y(x))}} - \frac{P(x,y(x))}{\sqrt{P(x)P(y(x))}}\right)^2 \\
&= 0,
\end{aligned}
$$

which means $\beta$ is an optimal solution of non-negative tuning.

# C ADDITIONAL EXPERIMENTS

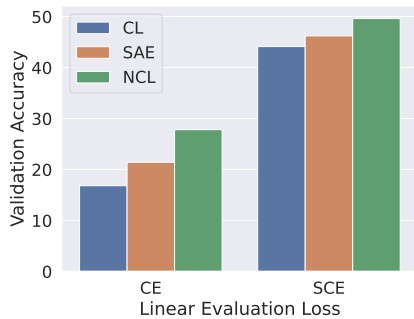

Figure 6: Linear probing performance with different evaluation losses on ImageNet-100 under 95% noise rates.

## C.1 COMBINATION WITH ROBUST LOSS

The previous results suggest that the monosemantic representations exhibit stronger robustness against label noise across various datasets. We note that there have been various studies to improve the robustness under label noise, such as applying robust loss functions (Van Rooyen et al., 2015; Ghosh et al., 2017), correcting training labels (Reed et al., 2014; Ma et al., 2018), and reweighting training samples (Chen et al., 2019; Han et al., 2018). However, the perspective in this paper is orthogonal to them. Taking the representative robust loss function Symmetric Cross Entropy (Wang et al., 2019b) as an example, we can obtain monosemantic representations as discussed above and then use the robust loss during the linear probing process. As shown in Figure 6, both the robust loss and enhancing feature monosemanticity can improve the robustness against label noise. Furthermore, the two methods are orthogonal, and combining them can further improve performance.

## C.2 EVALUATION OF MONOSEMANTICITY

We evaluate the monosemanticity of different models with current metrics. For vision models, we follow Wang et al. (2024) and adopt semantic consistency as the metric.The semantic consistency calculates the proportion of activated samples that belong to their most frequent class along each dimension. We evaluate the models in different cases, including the frozen models (Table 1, Figure 2) and the models fine-tuned with 20 % training samples and 20% noise rate (Figure 3).

As shown in Table 1, Non-negative Contrastive Learning (NCL), Sparse Autoencoder (SAE), and Non-negative tuning significantly improve the semantic consistency of trained models, which further supports our claim that attaining monosemanticity can enhance model robustness. Besides, we also note that the NCL obtains larger improvements on monosemanticity. which is consistent with the results that NCL performs better than SAE under noise.

## C.3 ADDITIONAL EVALUATION ON DIFFERENT NOISE RATES

Besides the noise rates in Table 2, we provide more detailed results for the 0-10% noise range results in the following table, with 3 runs and error bars. As shown in Table 4, we note that the monosemantic representations also show benefits with noise rates less equal to 10%, and the advantages rise with stronger noise rates.

## C.4 ADDITIONAL COMPARISON WITH NON-NEGATIVE CONTRASTIVE LEARNING

We note that there exist performance discrepancies bewteen our linear probing results with Wang et al. (2024). The discrepancies arise because we evaluate the linear probing accuracy on the features after the projector (an additional MLP after the backbone used in contrastive learning) while Wang et al. (2024) evaluate that on the features before the projector. We note that Wang et al. (2024) adds non-negative constraints and calculates the monosemanticity-related scores on the features after the

Table 4: Linear probing accuracy (%) under different noise rates (0-10%) on ImageNet-100 of polysemantic features (CL) and monosemantic features (NCL, SAE).

|      | 0 | 2 | 4 | 6 | 8 | 10 |
|------|---|---|---|---|---|----|
| CL   | **66.8±0.2** | 65.4±0.1 | 65.0±0.2 | 64.4±0.1 | 63.8±0.2 | 63.3±0.2 |
| NCL  | 66.8±0.2 | **66.2±0.3** | **66.1±0.4** | **66.1±0.1** | 65.5±0.3 | 65.5±0.1 |
| Gains | -0.0 | **+0.8** | **+1.1** | **+1.7** | **+1.7** | **+2.2** |
| SAE  | 66.1±0.2 | 65.9±0.1 | 65.9±0.1 | 65.7±0.3 | 65.5±0.3 | 65.2±0.2 |
| Gains | -0.6 | +0.5 | +0.9 | +1.3 | +1.7 | 1.9 |

Table 5: Linear probing accuracy (%) under different noise rates on ImageNet-100 of features before and after the projectors of models trained by CL and NCL.

| Noise Rate (%) | 0 | 30 | 60 | 90 |
|----------------|---|----|----|----|
| CL (w projector) | **66.8** | 60.1 | 54.9 | 34.4 |
| NCL (w projector) | 66.7 | **63.9** | **50.5** | **48.1** |
| CL (w/o projector) | 68.6 | 49.3 | 33.0 | 9.7 |
| NCL (w/o projector) | **69.5** | **50.6** | **33.7** | **9.8** |

projector. However, they calculate the linear evaluation accuracy of the features before the projector, which leads to a mismatch and we can not observe the relationship between monosemanticity and performance. Consequently, we calculate the performance of the features after the projector in this paper. Furthermore, we conduct additional experiments to evaluate the linear probing accuracy of the features before the projector.

As shown in Table 5, we observe that the improvements in accuracy do not increase with larger noise rates, which is reasonable as the features before the projector are not guaranteed to be monosemantic. Besides, we also note that the linear probing accuracy of the features after the projector performs better than the features before the projector under label noise, which further implies that we should use the features after the projector in this paper.

## C.5 ADDITIONAL EVALUATION ON LLMS

We also evaluate LLMs on additional benchmarks. To be specific, we respectively evaluate the original LLM models (Llama-2-7B-Chat) and the models fine-tuned on Dolly with LoRA and MonoRoLA on the MMLU benchmark.

As shown in Table 6, we observe that different from the security score in the main paper (Shield-Gemma Alignment Scores and Beavertails), the MMLU scores almost do not change during the fine-tuning process on Dolly with LoRA, which implies that the LLM abilities on the MMLU benchmark are mostly preserved during fine-tuning. Consequently, our method does not exhibit benefits as it is more like the clean accuracy of language models, which is consistent with our claims that monosemanticity shows benefits in the tasks related to model robustness.

## C.6 ABLATION STUDY ON SPARSE AUTOENCODERS

To evaluate how robust are these models to the choice of the number of activated features in SAEs, we evaluate the performance of SAE respectively with the number of activated features as 64, 128, 256, 512, and 1024.

Table 6: Evaluation of models tuned by LoRA and MonoLoRA on MMLU.

|          | Humantities | Other | Social Sciences | Stem | Average |
|----------|-------------|-------|-----------------|------|---------|
| Base     | 0.43 | 0.55 | 0.53 | 0.36 | 0.47 |
| LoRA     | 0.43 | 0.53 | 0.53 | 0.37 | 0.47 |
| MonoLoRA | 0.43 | 0.53 | 0.53 | 0.37 | 0.47 |

Table 7: Linear probing accuracy (%) of SAE with different numbers of activated features under different label noise rates on ImageNet-100.

| Noise Rate (%) | 64 | 128 | 256 | 512 | 1024 | w/o SAE |
|---|---|---|---|---|---|---|
| 0 | 64.2 | 64.7 | 66.1 | 66.4 | **66.8** | 66.8 |
| 30 | 59.3 | 59.6 | **60.6** | 60.4 | 60.4 | 60.1 |
| 60 | 58.3 | 58.2 | **58.6** | 55.7 | 55.2 | 54.9 |
| 90 | **45.9** | 45.9 | 45.7 | 43.1 | 38.9 | 34.5 |

As shown in Table 7, all the SAEs with different numbers of activated features exhibit benefits when there exist label noises. To be specific, when the activated features are small, the original performance will drop sharply when there exists no label noise. Meanwhile, when the label noise is aggressive, the small number of activated features will indicate stronger linear accuracy. Consequently, 256 is a sweet point that shows comparable performance in clean accuracy and exhibits benefits when there exists noise.

