# OpenReview forum: "Beyond Interpretability: The Gains of Feature Monosemanticity on Model Robustness"
_ICLR.cc/2025/Conference — ICLR 2025 Poster_

### Official Review · Reviewer_JMHa · 2024-10-21

**Soundness:** 3
**Presentation:** 2
**Contribution:** 2
**Rating:** 6
**Confidence:** 4

**Summary:**

In this work the authors show that inducing models to learn more monosemantic features (as opposed to neurons that learn multiple concepts, also known as polysemantic features) not only improves the interpretability of the model decisions, but it can also lead to more robust models.

The authors show results both in the vision and language domain. For the vision experiments they adopt two different techniques in order to induce monosemanticity: Non-Negative Contrastive Learning  (NCL, Wang et al 2024) which is a simple variation of SimCLR and it is supposed to create monosemantic representations, and Sparse Auto Encoders (SAE, Gao et al 2024) as a post-training intervention that produces embedding with higher monosemanticity. The authors use ReseNet-18 trained in a self-supervised fashion on CIFAR-100 and Imagenet-100. The authors show that when using either NCL to train the model or SAE (on the embedding of a pre trained SimCLR model) the accuracy of both models are comparable to that of the original model (that is usually polysemantic). However, when the dataset includes higher level of label noise, distribution shift (gaussian or uniform noise, real-data distribution shift), or the dataset is much smaller, the model that rely on monosemantic features remain more robust than the original model.

The authors also show some empirical analysis where they find that when training with a 90% of label noise, the dimension activated by the class with lower accuracy is polysemantic (i.e., is also activated by other classes), whereas the class with the higher dimension is monosemantic. Given this they conclude that noisy classifiers prefer monosemantic feature in practice.

For the language model the authors propose to modify the known Low Rank Adaptation mechanism (LoRA) by adding non-negative transformations (ReLU) that force the output of the LoRA layers to be more sparse (which is assumed to also induce monosemanticity). The authors fine-tune a Llama2 model on two different datasets (SST2 and Dolly) using the traditional LoRA and their proposed MonoLoRA, and show that while both models achieve comparable downstream task performance, MonoLoRA better preserves the model alignment (measured by the ShieldGemma and Beavertail alignment scores). In the case of the SST2 dataset the alignment becomes even better than that of the original model.

Lastly, the authors present an analysis using the super-position model (Elhage 2022b) that validates the results found on the vision models: in absence of noise models with polysemantic features show better performance, however, in the presence of noise the models with monosemantic features become more accurate than their polysemantic counterpart. Using a toy task (binary classification with a simple 2 dimensional embedding) the authors also show that when the label noise rate is grater than 25% the model with mono semantic features show a better linear separability between the two classes, again validating the empirical results found in the vision experiments on CIFAR-100 and ImageNet-100.

**Strengths:**

The paper investigate an aspect of monosemanticity that I am not aware that has been investigated before.

Among all the experiments presented I think the real-data distribution shift and the low percentage of training data represent important and realistic scenarios where monosemanticity could play an interesting role.

The paper is easy to read and I particularly liked the introduction where all the main findings are nicely summarized.

**Weaknesses:**

The main weakness I see in this work is related to the experimental setting. The methods used to induced monosemanticity are already known (with the exception of MonoLora) so I would have expected a more thorough evaluation. In particular these are the main points I would like to discuss with the authors: potentially unrealistic scenarios (e.g., label noise >= 25%),  some concerns with the quality of the empirical evaluation (e.g., baselines not matching state of the art results, lack of quantification of monosemanticity, lack of error bars, lack of evaluation of LLM capabilities). Addressing these points could improve the soundness and the contribution of this work. Let me elaborate more in detail each of those points.

- Potentially unrealistic scenarios. The experiments on labels noise explore a range of noise that goes from 0 to 90% which seems a very large and unrealistic amount of noise. I would consider exploring more in details (and with multiple runs) what happens in a more realistic range. The work from MIT “Pervasive Label Errors in Test Sets Destabilize Machine Learning Benchmarks” could be a useful guidance to know what is a realistic range. From that paper it seems the full ImageNet has about 3.4% of label errors, while CIFAR-100 has about 5.85% (from their table 1). The worst case is QuickDraw with 10.12% error rate. The average error rate from that table is 3.32%. Given those results, perhaps focusing on the range 0-10 would be more realistic and informative than 0-90.  I believe this paper would be very interesting (if results holds) if more space was dedicated to real-data distribution shifts (all imagenet test bed as mentioned below), training with low amount of data, and fine-tuning LLMs. All the experiments with label noise and other kind of noise would be interesting only if results showed that monosemanticity can help in realistic scenarios (e.g., label noise between 0% to 10%).

- Baselines not matching state of the art results. Even with a small scale model such as ResNet-18 (as opposed to state of the art work on self-supervised learning that use ResNet-50 - see one of my “minor” comments below about scaling to larger models) the results previously published by Wang et al. 2024 (table 3) show accuracies that are higher than the ones reported in this work for CIFAR-100 and ImageNet-100 (both for the original model and the NLC ones). Specifically: Wang et al. 2024 reports for CIFAR-100
Contrastive learning (with linear probing) 58.6% vs current work 54.5%
Non Contrastive learning (with linear probing) 59.7% vs current work 52.8%
Similar discrepancies can be found for ImageNet-100 with linear probing, and with both CIfar-100 and ImageNet-100 with full fine tuning using 100% of the datasets.
My concerns with these non-matching results are that on the one hand Wang et al. had already shown that NCL is better than traditional contrastive learning even in the original setting, and additionally, it undermine the credibility of the results shown. I would be great to comment on these discrepancies.

- Lack of quantification of “monosemanticity”. The paper rely on models with monosemantic features however the level of monosemanticity is never measured and compared with counterpart models. It would be great to quantitively show that indeed the model that are being produced do contain more monosemantic features. While NCL and SEA were already published and they might have shown their ability to increase monosemanticity, it would still be good to double check this effect. This is even more important with the proposed MonoLoRA since I suppose that no one has ever checked if in this case sparsity also leads to monosemanticity. Additionally an analysis of monosemanticity could help understanding more deeply the differences between NCL and SEA beyond accuracy (which of the two actually leads to more monosemantic representations?).

- None of the experiments show any error bar. It is always good to have multiple runs

- I found the explanation of the LLM experiments lacking necessary details in order to be understood properly. Without reading Appendix A3 (and other still missing details) this section is not self-explanatory. I would encourage rewriting this section with more details. For example, talk about SST2 and Dolly datasets: what kind of tasks are they for? On which metrics are they evaluated? What are the ShieldGemma-9B and Beaverstails-7B alignment scores?

- Additionally, related to the LLM evaluation, the authors mention overfitting as one of the issue with fine-tuning. If overfitting is the concern, I think that it would be good to show that the model has not lost some of its original capabilities (if they are still of interest). Usually this is shown by empirically measuring the change in perplexity, the MMLU benchmark and the common sense reasoning tasks. The current experiments only show the change in “alignment” (which should be better defined) but not other changes that could be affected by overfitting.


What follows are some suggestions on how to improve the writing, the clarity, and overall the quality of the manuscripts. While there are some questions in there they are not necessarily meant to be answered in the rebuttal but rather to be answered in a new version of the paper. None of the comments below are of major concern, however, addressing them would make this work stronger. Please focus the rebuttal on the questions related to the main concerns.

Writing:
- Since Wang et al. had already shown that there is no trade off to be paid (in their paper NCL works better than standard SimCLR) I would put the emphasis on showing that monosemanticity could be useful also in terms of robustness. Talking about the trade off seems in contradiction with Wang et al.’s results.
- Some statements in the abstract should be revised to avoid creating false expectations. For example the authors state “Our preliminary analysis suggests that monosemanticity, by promoting better separation of feature representations, leads to more robust decision boundaries.” This statement seems to be always “true” but from the results presented it seems true only in specific scenarios (high level of noise etc…).
- “More details can be found in Appendix A3“. When referring to the appendix consider offering more info about which details can be found in the appendix so the reader can decide if they want to read it or not. Are there more results? More background on the experiments? More info on the hyperparameters etc…
- Figure 4. The pictures in this figure refer to the model with induced monosemanticity or the original model? It could be interesting to compare this result with the “other” model.
- Figure 4. What does it mean “top activated” samples? What is the value for the highest accuracy group and the lowest accuracy group? Is the issue mono vs poly semanticity of could the issue simply be that the classes that perform poorly have not been properly learnt (that could be indicated by a low magnitude activation?).
- I would invite the authors to think how much the super-position section (both empirical and “theoretical”) add to the paper. I don’t find it particularly illuminating because it is still based on a very specific set of choices (hidden dimension size, task, etc…). Maybe this space could be used to explain more in details the LLM settings, or present experiments on larger scale, etc.. and this section could be moved to the appendix.
- In the introduction “Pretrained LLMs need to be carefully finetuned on small-scale language data for different purposes, e.g., instruction following and certain abilities (e.g., reasoning), while avoiding conflicting and forgetting.” What does “conflicting” means in this context? This should be clarified.
- All the experiments in 3.2.1 could be obtained on the NLC trained backbone or on the original SimCLR backbone. It is unclear what backbone was used both for the standard fine-tuning and the NNT fine-tuning. There are 4 possible combinations (standard SimCLR backbone with standard fine-tuning, standard SimCLR backbone with NNT, NCL backbone with standard fine-tuning and NCL backbone with NNT fine-tuning) it would be great to explicitly say which are being evaluated.
- Writing: I am assuming that the experiments in 3.2 are different than 3.1 because in 3.2 the full backbone model is fine-tuned while in 3.1 the backbone is frozen. However, this should be mentioned more precisely. Currently the paper says “Unlike linear probing we train classifiers on the pretrained representations without clipping the gradient of encoders”. However, clipping the gradient does not necessarily mean freezing the encoder. I would clarify this aspect.
- Suggestion related to Figure 2:  I’d suggest to include (maybe in the appendix) an example of a figure perturbed at the different levels used in the experiments so that the reader can appreciate better if that level of noise is realistic and in which scenario this degradation could happen.
- Page 5 mentions CIFAR-10 but according to the previous explanation I believe this should be CIFAR-100 (Typo)?
- The feature map analyzed in the case of the vision experiments are those before the logits? This detail was never explicitly mentioned (or I might have missed it).
- Throughout the paper there are some “concepts” that are never formally defined. For example, the authors often talk about a neuron being activated by single or multi set of concepts. What does it mean to be activated?

Other aspects (not critical but would improve the work):
- Another potential mismatch is with the SAE definition. In equation (2) the authors define the output for the encoder as  $z(x) = topK(W_{enc}(f(x)-b_{pre}) +b_{enc})$ and the output of the AE as $\hat{f}(x) = W_{dec}z(z) + b_{pre}$. However, Gao et al. uses a slightly different definition of the topK function which does not include the term $b_{enc}$, specifically, $z(x) = topK(W_{enc}(f(x)-b_{pre}))$. Is this a typo, are the two formulations equivalent, or is this an acltual difference compared to Gao et al. (in which case it would be great to motivate the change)?
- I believe the experiments with real-world distribution shift are very interesting and relevant. However the authors only reports results on 2 shifts (sketch and stylized). On the one hand the justification about why only these two out of the existing ones in the cited literature would be necessary. On the other hand it would be even better to include all of them and leverage advances in this field and extend the results to all imaginet-testbed shifts from Taori et al 2020 (“Measuring Robustness to Natural Distribution Shifts in Image Classification”). If authors could show a significant gain in robustness with respect to these distribution shifts the paper would be much stronger since some of these shifts are more realistic in my opinion than a 90% noise in the labels.
- Small scale experiments. It is often the case that when moving from a small scale setting (small model and small dataset) to a large scale many of the advantages observed are unfortunately lost. The authors use ResNet-18 with CIFAR-100 and ImageNet-100 but state of the art in self supervised learning is usually working with ResNet-50 and full ImageNet. I am not sure if the same results will transfer also to larger scale settings. For example, notice how in Figure 3(b) the gap between the accuracy of the polysemantic and the monosemantic models is reducing as you go towards 100% of training data. Had the authors used ResNet50 and full imageNet would you still observe such a difference? Perhaps this indicates that monosemantic features are very important in small scale settings but not in large ones?  This experiment is not crucial but without it I would assume that monosemantic is only useful in a low scale regime. If the authors want to extend the influence of monosemanticity on large scale setting this should be shown.
- I am not sure I would consider the “theoretical understanding” provided in section 4.3 actually theoretical. It seems to me that is a toy model with concrete values for different choices that can be analyzed manually. It has value but I would consider this still an empirical analysis.
- From the appendix the authors mentioned that the number of activated features for SAE was 256. How was this choice made and how robust are these models to the choice of this parameter?
- Why topK and not JumpRelu (Lieberum et al.)? It would be good to motivate this choice.
- In general the appendix contain most of the hyper-parameters used but I could not find the value of the Temperatures.

**Questions:**

On the unrealistic scenarios, could the authors:
1. Provide detailed results for the 0-10% noise range, with multiple runs and error bars.
2. Discuss whether the benefits of monosemanticity are still observed in this more realistic regime.
3. Consider emphasizing and expanding the experiments on real-world distribution shift experiments, as these may be more practically relevant.

On the discrepancies with Wang et al. 2024 results for CIFAR-100 and ImageNet-100, could the authors:
1. Provide a detailed explanation for the discrepancies.
2. Verify their implementation of NCL to ensure it matches the original method.
3. If possible, reproduce the exact settings from Wang et al. to confirm whether the discrepancy persists.

On the quantification of “monosemanticity”, could the authors:
1. Measure and report a metric of monosemanticity (e.g., semantic consistency as used in Wang et al.) for all models and methods, including MonoLoRA.
2. Provide a comparative analysis of monosemanticity levels between NCL, SAE, and standard models.
3. Investigate the relationship between sparsity and monosemanticity in MonoLoRA.

On the lack of error bars. It is important to repeat the experiments (especially when stochastic processes such sampling are involved) and provide mean and standard deviation. Are the results shown statistically significant?

On the lack of evaluation of LLMs: would it make sense to show that the LLM abilities are better preserved with monosemanticity by evaluating the MMLU benchmark?

Further questions
- In the abstract the authors called these analysis “Preliminary” could you clarify which of the results you consider preliminary and what you would need to consider them definitive?
- The end of the intro reads “Theoretically, as a preliminary step, we compare polysemantic and monosemantic features under a toy model proposed in Elhage et al. (2022b). The theory suggests that because monosemantic features have better separation of features, they are less prone to overfitting to noise, leading to more robust decision boundaries compared to polysemantic features.”. Is this what the theoretical analysis concluded? If so, shouldn’t the benefit be evident also in noise-free scenarios? I believe that what it is shown in section 4.3 matches this statement only when in presence of a large amount of noise (>=25%). Or have I misunderstood what the authors mean?

---

> ### Author Response · Authors · 2024-11-25
> **Response to Reviewer JMHa (1/3)**
>
> Thanks very much for your constructive and detailed comments. Below, we will address your main concerts point by point.
>
> ---
>
> Q1. Additional evaluation in more realistic scenarios.
>
> A1. Thanks for your suggestions! We provide more detailed results for the 0-10% noise range results in the following table, with 3 runs and error bars.
>
> *Table1. Linear probing accuracy (%) under different noise rates (0-10%) on ImageNet-100 of polysemantic features (CL) and monosemantic features (NCL, SAE).*
>
> |  | 0 | 2 | 4 | 6 | 8 | 10 |
> | --- | --- | --- | --- | --- | --- | --- |
> | CL | **66.8$\pm$0.2** | 65.4$\pm$0.1 | 65.0$\pm$0.2 | 64.4$\pm$0.1 | 63.8$\pm$0.2 | 63.3$\pm$0.2 |
> | NCL |  **66.8$\pm$0.2**  | **66.2$\pm$0.3** | **66.1$\pm$0.4** | **66.1$\pm$0.1** | **65.5$\pm$0.3** | **65.5$\pm$0.1** |
> | Gains | -0.0 | **+0.8** | **+1.1** | **+1.7** | **+1.7** | **+2.2** |
> | SAE | 66.1$\pm$0.2 | 65.9$\pm$0.1 | 65.9$\pm$0.1 | 65.7$\pm$0.3 | 65.5$\pm$0.3 | 65.2$\pm$0.2 |
> | Gains | -0.6 | +0.5 | +0.9 | +1.3 | +1.7 | +1.9 |
>
> Besides, we follow your suggestions and conduct more experiments on real-world distribution shifts. As the link of the dataset in [1] is not accessible currently, we use another OOD benchmark [2], which provides about 100 different distribution shifts including shot noise, impulse noise, defocus blur, etc. We calculate the linear probing accuracy of models trained with contrastive learning, non-negative contrastive learning, and sparse autoencoder under different shifts and calculate the mean accuracy.
>
> *Table2. The average linear probing accuracy on ImageNet-C benchmark.*
>
> |  | CL | SAE | NCL |
> | --- | --- | --- | --- |
> | Accuracy | 31.7$\pm$0.3 | 32.5$\pm$0.4 | **33.6$\pm$0.3** |
>
> As shown in the above table, enhancing monosemanticity obtains superior performance under different kinds of real-world shifts, which further verifies our claims in a more realistic scenario.
>
> ---
>
> Q2. The discrepancies with Wang et al. 2024 results for CIFAR-100 and ImageNet-100.
>
> In fact, the discrepancies arise because we evaluate the linear probing accuracy on the features after the projector (an additional MLP after the backbone used in contrastive learning) while [1] evaluate that on the features before the projector.  **We note that [1] adds non-negative constraints and calculates the monosemanticity-related scores on the features after the projector. However, they calculate the linear evaluation accuracy of the features before the projector,** which leads to a mismatch and we can not observe the relationship between monosemanticity and performance. Consequently, we calculate the performance of the features after the projector in this paper. Furthermore, we conduct additional experiments to evaluate the linear probing accuracy of the features before the projector.
>
> *Table3. Linear probing accuracy (%) under different noise rates on ImageNet-100 of features before and after the projectors of models trained by CL and NCL.*
>
> | Noise Rate (%) | 0 | 30 | 60 | 90 |
> | --- | --- | --- | --- | --- |
> | CL (w projector) | 66.8 | 60.1 | 54.9 | 34.4 |
> | NCL (w projector) | 66.7 | **63.9** | 50.5 | 48.1 |
> | CL (w/o projector) | 68.6 | 49.3 | 33.0 | 9.7 |
> | NCL (w/o projector) | **69.5** | 50.6 | 33.7 | 9.8 |
>
> As shown in the above table, we observe that the improvements in accuracy do not increase with larger noise rates, which is reasonable as the features before the projector are not guaranteed to be monosemantic. Besides, we also note that the linear probing accuracy of the features after the projector performs better than the features before the projector under label noise, which further implies that we should use the features after the projector in our paper.

---

> > ### Comment · Reviewer_JMHa · 2024-11-25
> > **Re: Response to Reviewer JMHa (1/3)**
> >
> > Thank you for providing all these new results and answers.
> >
> > I believe these results strengthen the paper. It would be great if you could also add in case of acceptance the results on distribution shift in [1] (I understand that currently this is not possible).
> >
> > I would include (maybe in the appendix) the explanation about the differences in the evaluation setting with respect to Wang et al 2024.

---

> ### Author Response · Authors · 2024-11-25
> **Response to Reviewer JMHa (2/3)**
>
> Q3. The evaluation of monosemanticity.
>
> A3. Thanks for your suggestions, we evaluate the monosemanticity of different models with current metrics. For vision models, we follow [1] and use semantic consistency as the metric. The semantic consistency calculates the proportion of activated samples that belong to their most frequent class along each dimension. We evaluate the models in different cases, including the frozen models (Table 1, Figure 2) and the models fine-tuned with 20 % training samples and 20% noise rate (Figure 3).
>
> *Table 4. Semantic consistency of different models.*
>
> |  | Poly (CL) | Mono (NCL) | Mono (SAE) |
> | --- | --- | --- | --- |
> | Linear Probing on CIFAR-100 | 1.0 | **8.2** | 3.1 |
> | Linear Probing on Imagenet-100 | 1.0 | **12.3** | 7.2 |
> |  | Poly CE) | Mono (NCE) |  |
> | Fine-tuning with 20% Samples | 2.1 | **18.4** |  |
> | Fine-tuning with 20% Noisy Lables | 3.4 | **20.6** |  |
>
>  As shown in the above table, Non-negative Contrastive Learning, Sparse Autoencoder and Non-negative tuning significantly improve the semantic consistency of trained models, which further supports our claims that attaining monosemanticity can enhance the model robustness.  Besides, we also note that the improvement of monosemanticity of NCL is larger than SAE, which aligns with the results that NCL performs better than SAE under different noises.
>
> For language models, as we have no access to labels, current works [2,3] usually use sparsity as a surrogate metric to evaluate the monosmenaticity. Specifically, we evaluate the sparsity (zero value ratio) of the intermediate activations of the LoRA and MonoLoRA models, which is the intrinsic sparsity of the LoRA module.
>
> *Table 5. Evaluation of monosemanticity on LoRA and MonoLoRA with Llama-2-7B-Chat on SST2 and Dolly.*
>
> |  |  | Alignment Sparsity | Task Sparsity |
> | --- | --- | --- | --- |
> | SST2 | LoRA | 0 | 0 |
> |  | MonoLoRA | 45.54 | 36.71 |
> | Dolly | LoRA | 0 | 0 |
> |  | MonoLoRA | 38.69 | 40.00 |
>
>  As shown in the table above, MonoLoRA significantly improves the sparsity of trained models, which indicates that our method also enhances the monosemanticity of large language models.
>
> **Reference**
>
> [1] Wang, Yifei, et al. "Non-negative Contrastive Learning." ICLR, 2024.
>
> [2] Yan, Hanqi, et al. "Encourage or Inhibit Monosemanticity? Revisit Monosemanticity from a Feature Decorrelation Perspective." EMNLP 2024.
>
> [3] Gurnee, Wes, et al. "Finding neurons in a haystack: Case studies with sparse probing." arXiv preprint arXiv:2305.01610 (2023).
>
> ---
>
> Q4.On the lack of error bars. It is important to repeat the experiments (especially when stochastic processes such sampling are involved) and provide mean and standard deviation. Are the results shown statistically significant?
>
> A4. Thanks for your suggestions, we repeat the experiments on Table 1, Figure 2 and Figure 3 three times and calculate the standard deviation. As shown in Table 1, Figure, and Figure 3 in the new revision, the results further verify that the improvements are statistically significant.
>
> ---
>
> Q5. I found the explanation of the LLM experiments lacking necessary details in order to be understood properly.
>
> A5. Thanks for your suggestions and we have rewritten this section with more details (Section 3.3.2 and Appendix A.1). For example, we have supplemented the details about SST2 and Dolly datasets and we explained the ShieldGemma-9B and Beaverstails-7B alignment scores.
>
> ---
>
> Q6. On the lack of evaluation of LLMs: would it make sense to show that the LLM abilities are better preserved with monosemanticity by evaluating the MMLU benchmark?
>
> A6. Thanks for your suggestions, we respectively evaluate the original LLM models (Llama-2-7B-Chat) and the models fine-tuned on Dolly with LoRA and MonoRoLA on the MMLU benchmark.
>
> *Table 6. Evaluation of models tuned by LoRA and MonoLoRA on MMLU.*
>
> |  | Humantities | Other | Social Sciences | Stem | Average |
> | --- | --- | --- | --- | --- | --- |
> | Base | 0.43 | 0.55 | 0.53 | 0.36 | 0.47 |
> | LoRA | 0.43 | 0.53 | 0.53 | 0.37 | 0.47 |
> | MonoLoRA | 0.43 | 0.53 | 0.53 | 0.37 | 0.47 |
>
> As shown in the table above, we observe that different from the security score in the main paper (ShieldGemma Alignment Scores and Beavertails), the MMLU scores almost do not change during the fine-tuning process on Dolly with LoRA, which implies that the LLM abilities on the MMLU benchmark are mostly preserved during fine-tuning. Consequently, our method does not exhibit benefits as it is more like the clean accuracy of language models, which is consistent with our claims that monosemanticity shows benefits in the tasks related to model robustness.

---

> > ### Comment · Reviewer_JMHa · 2024-11-25
> > **Re: Response to Reviewer JMHa (2/3)**
> >
> > Thanks, again I think these results are important for strengthening your work. Please consider including them if you haven't already done so.

---

> ### Author Response · Authors · 2024-11-25
> **Response to Reviewer JMHa (3/3)**
>
> Q7.  In the abstract the authors called these analysis “Preliminary” could you clarify which of the results you consider preliminary and what you would need to consider them definitive?
>
> A7. We call these analyses preliminary because we establish theoretical analysis in toy model settings while the empirical results show the benefits of monosemanticity exhibited across various real-world scenarios, including linear probing and fine-tuning with label noise, distribution shifts, etc. In future work, establishing further theoretical analysis in realistic scenarios would be a valuable direction to verify the benefits of monosemanticity and inspire more researchers to unlock the benefits of monosemanticity beyond interpretability.
>
> ---
>
> Q8. The end of the intro reads “The theory suggests that because monosemantic features have better separation of features, they are less prone to overfitting to noise, leading to more robust decision boundaries compared to polysemantic features.”. Is this what the theoretical analysis concluded? If so, shouldn’t the benefit be evident also in noise-free scenarios? I believe that what it is shown in section 4.3 matches this statement only when in presence of a large amount of noise (>=25%). Or have I misunderstood what the authors mean?
>
> A8. Indeed, we want to deliver that the theory implies monosemantic features have better separation of features under noise. We understand that the original sentence would lead to misunderstandings. Consequently, we have adjusted it to “The theory suggests that because monosemantic features have better separation of features under large noise rates (≥ 25%), they are less prone to overfitting to noise, leading to more robust decision boundaries compared to polysemantic features.”.
>
> ---
>
> Q9. definition of SAE
>
> A9. Thanks for pointing it out and it is a typo. We have fixed it in the new revision.
>
> ---
>
> Q10.  From the appendix the authors mentioned that the number of activated features for SAE was 256. How was this choice made and how robust are these models to the choice of this parameter?
>
> A10. That is a good question! In the following table, we evaluate the performance of SAE respectively with the number of activated features as 16, 128, 256, 512, and 1024.
>
> *Table 7. Linear probing accuracy (%) of SAE with different numbers of activated features under different label noise rates on ImageNet-100.*
>
> | Noise Rate (%) | 64 | 128 | 256 | 512 | 1024 | w/o SAE |
> | --- | --- | --- | --- | --- | --- | --- |
> | 0 | 64.2 | 64.7 | 66.1 | 66.4 | **66.8**| 66.8 |
> | 30 | 59.3 | 59.6 | **60.6** | 60.4 | 60.4 | 60.1 |
> | 60 | 58.3 | 58.2 | **58.6** | 55.7 | 55.2 | 54.9 |
> | 90 | **45.9** | 45.9 | 45.7 | 43.1 | 38.9 | 34.5 |
>
> As shown in the table above,  all the SAEs with different numbers of activated features exhibit benefits when there exist label noises. To be specific, when the activated features are small, the original performance will drop sharply when there exists no label noise. Meanwhile, when the label noise is aggressive, the small number of activated features will indicate stronger linear accuracy. Consequently, 256 is a sweet point that shows comparable performance in clean accuracy and exhibits benefits when there exists noise.
>
> ---
>
> Q11. In general, the appendix contains most of the hyper-parameters used but I could not find the value of the Temperatures.
>
> A11. We have added the settings of temperatures in Appendix A.1.
>
> ---
>
> Hope our explanations and newly added experiments above could address your concerns. We are looking forward to your reply and please let us know if there is more to clarify.

---

> > ### Comment · Reviewer_JMHa · 2024-11-25
> > **Re: Response to Reviewer JMHa (3/3)**
> >
> > Thank you again for all the explanation.
> >
> > I believe that all the new results and clarification if included in the manuscript will strengthen this work considerably. I am going to upgrade by score and recommendation accordingly.

---

> > > ### Author Response · Authors · 2024-11-25
> > > **Further Response to Reviewer JMHa**
> > >
> > > Dear Reviewer JMHa,
> > >
> > > Thanks for your responses and for improving the scores! We have updated the revision and added these results and explanations in the manuscript, including 1) evaluation results of real-world distribution shifts benchmark ImageNet-C in Section 3, Figure 2(c); 2) evaluation results of 0-10% noise rates in Appendix C.3; 3) empirical results and explanations for the differences in the linear probing setting with respect to Wang et al 2024 in Appendix C.4; 4) evaluation results of monosemanticity in Appendix C.2; 5) evaluation results of MMLU benchmark in Appendix C.5; 6) evaluation results of different numbers of activated features in SAEs in Appendix C.6. Thanks again and we would be happy to address any remaining concerns.
> > >
> > > Best, Authors

---

> ### Author Response · Authors · 2024-12-02
> **Your further inputs are greatly appreciated. Only 24 hours left.**
>
> Dear Reviewer JMHa,
>
> Following your suggestions, we have included additional results and discussions in the revision of the manuscript. We were hoping to hear your feedback on them.
>
> As there are only 24 hours left for the reviewer and author discussions and we understand that everyone has a tight schedule, we kindly wanted to send a gentle reminder to ensure that our revision sufficiently addresses your concerns or if there are further aspects we need to clarify.
>
> If you could find the time to provide your thoughts on our revision, we would greatly appreciate it.
>
> Best, Authors

---

> > ### Comment · Reviewer_JMHa · 2024-12-02
> >
> > Thank you for updating your manuscript. At this time I do not have any further comment.

---

### Official Review · Reviewer_qpFh · 2024-11-02

**Soundness:** 3
**Presentation:** 3
**Contribution:** 2
**Rating:** 5
**Confidence:** 3

**Summary:**

The work tackles the problem of polysemantic neurone, i.e., when in the last layer of a neural network individual neurones are activated by multiple concepts. The authors state to break the accuracy-interpretability tradeoff and show that monosemantic neurones can provide more robust models as well as gain of accuracy. The authors also present theoretical calculations connected to monosemantic and polysemantic features.

**Strengths:**

The observations made in the manuscript regarding the monosemanticity seem original.
Quality of the research cannot be disputed.

**Weaknesses:**

1. Reading and understanding the manuscript relatively well requires a specific set/baggage of knowledge to have been acquired, because the manuscript is full of branch-specific references and acronyms. This makes the manuscript difficultly accessible for general public.

Perhaps as a possible improvement, the author(s) could prepare more intuitive writing for the body of the manuscript but shift (more difficult to understand) acronymed details to appendix.

2. I do not find a clear methodological contribution, rather an empirical finding.

3. What authors call Theorems 4.1 and 4.2 are (mathematically speaking) examples, not theorems.

**Questions:**

I have counted 4 data sets (for different tasks), is it really enough to make any general statement? If yes, the authors should provide an argument why? E.g., can the performance differences from Table 1 be called statistically significant? If yes, with which p-value(s)?

---

> ### Author Response · Authors · 2024-11-25
> **Response to Reviewer qpFh (1/3)**
>
> Thanks for careful reading and appreciating the quality of our work. We will summarize and address the main weakness points and the questions you mentioned.
>
> ---
>
> Q1. Reading and understanding the manuscript relatively well requires a specific set/baggage of knowledge to have been acquired. As a possible improvement, the author(s) could prepare more intuitive writing for the body of the manuscript but shift (more difficult to understand) acronymed details to the appendix.
>
> A1. Thanks for your suggestions and we have adjusted our writing in the revision. To be specific, we have 1) added more detailed and intuitive explanations of the “accuracy-interpretability” trade-off in Lines 41-45; 2) explained the acronyms in Line 89 (LoRA) with intuitive writing; 3) revised the caption in FIgure1 and replaced acronyms (CL, NCL, SAE); 4) replaced the acronyms in Line 204 (MSE) with intuitive writing; 5) replaced the acronyms in Line 318 (LLM) with explanations. If you encounter any other parts that are difficult to understand, please let us know, and we will address them in the revision.

---

> ### Author Response · Authors · 2024-11-25
> **Response to Reviewer qpFh (2/3)**
>
> Q2. Do not find a clear methodological contribution.
>
> A2. We understand your concern that our use of existing methods to examine the relationship between monosemanticity and robustness in our experiments might limit the novelty of our approach. However, we believe our paper offers clear contributions to the community and provides fresh insights into the topic of monosemanticity. Specifically, our work introduces several key distinctions from previous research.
>
> 1.  **We adopt the techniques that promote monosemanticity from an entirely new perspective.** The existing works about monosemanticity focus on feature interpretability. For example, Sparse Autenocders [1] and Non-negative Contrastive Learning [2] respectively discuss the interpretability in Language and Vision domains and propose different interoperability scores. To the best of our knowledge, this is the first study to investigate the relationship between monosemanticity and robustness. We note that adopting well-researched techniques in a new area can achieve impressive performance and provide novel insights. For example, another famous technique in the robustness area,  i.e., adversarial training, adopts min-max optimization in the robustness area and becomes the widely-used paradigm. Consequently, we believe that revealing learning benefits beyond interpretability as a new direction is non-trivial.
> 2. **We challenge the interpretability-accuracy trade-off in realistic and widely appearing scenarios.** The existing work observes that monosemanticity usually hurts the model performance. For example [1] observed that the sparse autoencoders significantly enhance the monosemanticity of language models while obtaining inferior performance in downstream prediction tasks. [3] show that monosemantic representations indicate lower model performance in reconstruction tasks with a toy model. [4] exhibit a negative correlation between expressive power and monosemanticity. In our papers, we find that when considering the robustness-related tasks, instead of hurting the performance, monosmenaticity significantly improves the accuracy. We believe this is a novel finding as it challenges the common belief and can push the area forward.
> 3. **We establish a new theoretical perspective of the monosemanticity.** **We follow the toy model settings [3] in the theoretical analysis. However, we note that the original analysis in [3] mainly focuses on how to obtain monosmeantic representations. e.g., they analyze the learning dynamic of monosmenatic features and the relationship between monosemanticity and the superposition hypothesis. In our paper, we theoretically analyze the performance of monosmeantic representations in downstream tasks, which provides a new perspective to theoretically understand the benefits of monosemantic features.
> 4. **We propose new implementations to enhance monomsenaticity in large language models.** We note that although NCL and SAE are existing techniques, the implementation of enhancing monosemanticity during the fine-tuning process of LLMS, i.e., MonoLoRA, is proposed in this paper, which shows impressive performance as in Table 4.
> 5. **We think that our methodological contribution also lies in proposing a new paradigm to improve the model robustness,** i.e., we design a pipeline where we first obtain monosemantic features by modifying pretraining algorithms and then enhance monosemanticity during the fine-tuning process to improve model robustness. When implementing the pipeline, we use some existing methods (NCL and SAE) as they have exhibited satisfactory performance.
>
> We have added a brief summary of this discussion in **Section 2**.
>
> Reference
>
> [1] Gao, Leo, et al. "Scaling and evaluating sparse autoencoders." arXiv preprint arXiv:2406.04093 (2024).
>
> [2] Wang, Yifei, et al. "Non-negative Contrastive Learning." *arXiv preprint arXiv:2403.12459* (2024).
>
> [3] Elhage, Nelson, et al. "Toy models of superposition." arXiv preprint arXiv:2209.10652 (2022).
>
> [4] Hoagy Cunningham, Aidan Ewart, Logan Riggs, Robert Huben, and Lee Sharkey. "Sparse autoen
> coders find highly interpretable features in language models." ICLR, 2024.

---

> ### Author Response · Authors · 2024-11-25
> **Response to Reviewer qpFh (3/3)**
>
> Q3.  What authors call Theorems 4.1 and 4.2 are (mathematically speaking) examples, not theorems.
>
> A3. Indeed, it is a common solution to establish theoretical analysis in a toy model with specific settings (e.g., the input data distributions, the specific network architectures, etc) for the simplicity of analysis. For example, [1] and [2] respectively analyze the learning process of different representation learning paradigms in a toy model setting. [3], [4] analyze different properties of monosemanticity in a toy model setting.
>
> On the other hand, we understand that the original analysis is a little too specific. To further generalize our analysis, we adjust our theoretical toy example and use a more general form. Specifically, we replace the definition of the polysemantic feature v=x1−x2 with w1x1+w2x2 in the theoretical analysis.  As shown in Theorem B.13, we note that we obtain a similar result, which still implies that the monosemantic representations indicate more robust performance under label noise and input noise.
>
> [1] Wen, Zixin, and Yuanzhi Li. Toward understanding the feature learning process of self-supervised contrastive learning. ICML, 2021.
>
> [2] Tian, Yuandong, Xinlei Chen, and Surya Ganguli. Understanding self-supervised learning dynamics without contrastive pairs. ICML 2021
>
> [3]Elhage, Nelson, et al. Toy models of superposition. arXiv preprint arXiv:2209.10652 (2022).
>
> [4] Lyu, Kaifeng, et al. Gradient descent on two-layer nets: Margin maximization and simplicity bias. NeurIPS 2021.
>
> ---
>
> Q4. I have counted 4 data sets (for different tasks), is it really enough to make any general statement?  Can the performance differences from Table 1 be called statistically significant?
>
> A4. For the number of datasets, we add additional experiments on CIFAR-10 and STL-10. We take linear probing under label noise as an example. To be specific, we respectively train ResNet-18 with CL and NCL and then evaluate the linear probing accuracy under different noise rates.
>
> *Table 1. Linear probing accuracy (%) of polysemantic representations (CL) and monosemantic representations (NCL) on CIFAR-10 and STL-10 under different noise rates (%).*
>
> |  |  | 0 | 20 | 40 | 60 | 80 |
> | --- | --- | --- | --- | --- | --- | --- |
> | STL-10 | CL | **63.9** | 41.9 | 34.4 | 29.4 | 14.7 |
> |  | NCL | 61.5 | **58.2** | **49.2** | **35.2** | **18.5** |
> |  | Gains | -2.4 | **+16.3** | **+14.8** | **+5.8** | **+3.8** |
> | CIFAR-10 | CL | **86.9** | 85.0 | 84.5 | 84.1 | 81.2 |
> |  | NCL | 86.7 | **86.2** | **85.8** | **85.4** | **82.7** |
> |  | Gains | -0.2 | **+1.2** | **+1.3** | **+1.3** | **+1.5** |
>
> As shown in the table above, enhancing monosemanticity also improves the linear probing performance on STL-10 and CIFAR-10 under different noise rates, which further verifies our claims that enhancing monosemanticity can improve the performance of model robustness.
>
> Besides, following your suggestions, we repeat the experiments 3 times and calculate the p-values to examine whether the performance differences from Table 1 are statistically significant.
>
> *Table 2. Linear probing accuracy (%) of polysemantic representations (CL) and monosemantic representations (NCL, SAE)  on CIFAR-100 and ImageNet-100 under different noise rates (%) and p-values of t-test between them.*
>
> | Dataset | Features |  | 0 | 10 | 30 | 60 | 90 |
> | --- | --- | --- | --- | --- | --- | --- | --- |
> | CIFAR-100 | Poly | Accuracy | 54.6$\pm$0.2 | 53.2$\pm$0.2 | 52.1$\pm$0.4 | 49.5$\pm$0.2 | 35.4$\pm$0.3 |
> |  | Mono(SAE) | Accuracy | 54.5$\pm$0.1 | 53.9$\pm$0.2 | 53.0$\pm$0.3 | 50.6$\pm$0.4 | 38.0$\pm$0.1 |
> |  |  | P-Value | $3*10^{-1}$ | $7*10^{-3}$ | $2*10^{-2} $| $1*10^{-2} $|$ 7*10^{-5}$ |
> |  | Mono(NCL) | Accuracy | 52.8$\pm$0.4 | 54.2$\pm$0.3 | 52.7$\pm$0.2 | 51.5$\pm$0.2 | 45.0$\pm$0.2 |
> |  |  | P-Value | $2*10^{-3}$ |$ 5*10^{-3}$ | $7*10^{-2}$ | $1*10^{-4} $| $1*10^{-6} $|
> | ImageNet-100 | Poly | Accuracy | 66.8$\pm$0.2 | 63.3$\pm$0.2 | 60.1$\pm$0.2 | 54.9$\pm$0.4 | 34.5$\pm$0.2 |
> |  | Mono(SAE) | Accuracy | 66.1$\pm$0.2 | 65.2$\pm$0.2 | 60.7$\pm$0.2 | 58.6$\pm$0.1 | 45.7$\pm$0.2 |
> |  |  | P-Value | $5*10^{-3}$ | $2*10^{-4}$ | $2*10^{-2}$ | $1*10^{-4}$ | $1*10^{-7}$ |
> |  | Mono(NCL) | Accuracy | 66.8$\pm$0.2 | 65.5$\pm$0.1 | 63.8$\pm$0.2 | 60.8$\pm$0.2 | 48.1$\pm$0.2 |
> |  |  | P-Value |$ 6*10^{-1}$ | $4*10^{-5}$ | $3*10^{-5}$ | $2*{10}^{-5} $|$ 7*10^{-8}$|
>
> As shown in the tables, the p-values are quite small (≤ 0.05) when there exists label noise, which verifies that the performance differences from Table 1 are statistically significant.
>
> ---
>
> Hope our discussion and newly added experiments could address your concerns. We are looking forward to your reply and please let us know if there is more to clarify.

---

> ### Author Response · Authors · 2024-11-27
>
> Dear Reviewer qpFh,
>
> We have carefully prepared a detailed response to address each of your questions. Would you please take a look and let us know whether you find it satisfactory?
>
> We note that Reviewer JMHa has appreciated our response and raised the score beyond the acceptance bar. We also respectfully suggest that you could re-evaluate our work with the updated explanations and results.
>
> Thanks! Have a great day!
>
> Authors

---

> ### Author Response · Authors · 2024-12-02
> **Your further inputs are greatly appreciated. Only 24 hours left.**
>
> Dear Reviewer qpFh,
>
> For your raised questions, we prepared a detailed response to address your concerns. We were hoping to hear your feedback on them.
>
> As there are only 24 hours left for the reviewer and author discussions and we understand that everyone has a tight schedule, we kindly wanted to send a gentle reminder to ensure that our response sufficiently addressed your concerns or if there are further aspects we need to clarify.
>
> If you could find the time to provide your thoughts on our response, we would greatly appreciate it.
>
> Best, Authors

---

### Official Review · Reviewer_u9M1 · 2024-11-03

**Soundness:** 2
**Presentation:** 4
**Contribution:** 3
**Rating:** 6
**Confidence:** 3

**Summary:**

The  authors  propose  that  monosemanticity,  where  neurons  represent  distinct  and  consistent  semantics,  can  improve  interpretability  without  compromising  accuracy.  The  authors  demonstrate  that  models  using  monosemantic  features  perform  better  in  three  robust  learning  scenarios.  The  study  provides  empirical  and  theoretical  evidence  that  monosemanticity  leads  to  more  robust  decision  boundaries,  suggesting  a link  between  interpretability  and  robustness.

**Strengths:**

1. The paper offers a valuable and insightful perspective on the benefits of feature monosemanticity.

2. The study presents a robust and diverse set of evidence and experiments from various learning scenarios.

3. The paper is well-organized, with a clear and logical structure that makes it easy to follow.

**Weaknesses:**

1. The paper utilizes existing sparse and non-negative constraints in three tasks, which are well-researched areas. While the application of these constraints to monosemanticity is interesting, the overall novelty is somewhat limited.

2. The paper does not provide a thorough explanation for why non-negative constraints are generalized across all three tasks. It is unclear whether the same form of non-negative constraints is suitable for all tasks or if specific forms are needed for different scenarios. More detailed discussion and additional experiments would help to clarify this point and strengthen the argument.

3. Abbreviations such as NCL and SAE, introduced in line 69, should be defined at their first occurrence. It well be helpful for readers to follow.

**Questions:**

See weakness.

---

> ### Author Response · Authors · 2024-11-25
> **Response to Reviewer u9M1 (1/2)**
>
> Thanks for your detailed reading. We address your main concerns (the novelty and the explanation for why non-negative constraints can be generalized) as follows.
>
> ---
>
> Q1.Utilizing existing methods, the overall novelty is somewhat limited
>
> A1. We understand your concern that our use of existing methods to examine the relationship between monosemanticity and robustness in our experiments might limit the novelty of our approach. However, we believe our paper offers clear contributions to the community and provides fresh insights into the topic of monosemanticity. Specifically, our work introduces several key distinctions from previous research.
>
> 1.  **We adopt the techniques that promote monosemanticity from an entirely new perspective.** The existing works about monosemanticity focus on feature interpretability. For example, Sparse Autenocders [1] and Non-negative Contrastive Learning [2] respectively discuss the interpretability in Language and Vision domains and propose different interoperability scores. To the best of our knowledge, this is the first study to investigate the relationship between monosemanticity and robustness. We note that adopting well-researched techniques in a new area can achieve impressive performance and provide novel insights. For example, another famous technique in the robustness area,  i.e., adversarial training, adopts min-max optimization into the robustness area and becomes the widely-used paradigm. Consequently, we believe that revealing the learning benefits of monosemanticity beyond interpretability as a new direction is non-trivial.
> 2. **We challenge the interpretability-accuracy trade-off in realistic and widely appearing scenarios.** The existing work observes that monosemanticity usually hurts the model performance. For example [1] observed that the sparse autoencoders significantly enhance the monosemanticity of language models while obtaining inferior performance in downstream prediction tasks. [3] show that monosemantic representations indicate lower model performance in reconstruction tasks with a toy model. [4] exhibit a negative correlation between expressive power and monosemanticity. In our papers, we find that when considering the robustness-related tasks, instead of hurting the performance, monosmenaticity significantly improves the accuracy. We believe this is a novel finding as it challenges the common belief and can push the area forward.
> 3. **We establish a new theoretical perspective of the monosemanticity.** **We follow the toy model settings [3] in the theoretical analysis. However, we note that the original analysis in [3] mainly focuses on how to obtain monosmeantic representations. e.g., they analyze the learning dynamic of monosmenatic features and the relationship between monosemanticity and the superposition hypothesis. In our paper, we theoretically analyze the performance of monosmeantic representations in downstream tasks, which provides a new perspective to theoretically understand the benefits of monosemantic features.
> 4. **We propose new implementations to enhance monomsenaticity in large language models.** We note that although NCL and SAE are existing techniques, the implementation of enhancing monosemanticity during the fine-tuning process of LLMS, i.e., MonoLoRA, is proposed in this paper, which shows impressive performance as in Table 4.
> 5. **We think that our methodological contribution also lies in proposing a new paradigm to improve the model robustness,** i.e., we design a pipeline where we first obtain monosemantic features by modifying pretraining algorithms and then enhance monosemanticity during the fine-tuning process to improve model robustness. When implementing the pipeline, we use some existing methods (NCL and SAE) as they have exhibited satisfactory performance.
>
> We have added a brief summary of this discussion in **Section 2**.
>
> Reference
>
> [1] Gao, Leo, et al. "Scaling and evaluating sparse autoencoders." arXiv preprint arXiv:2406.04093 (2024).
>
> [2] Wang, Yifei, et al. "Non-negative Contrastive Learning." ICLR, 2024.
>
> [3] Elhage, Nelson, et al. "Toy models of superposition." arXiv preprint arXiv:2209.10652 (2022).
>
> [4] Hoagy Cunningham, Aidan Ewart, Logan Riggs, Robert Huben, and Lee Sharkey. "Sparse autoen
> coders find highly interpretable features in language models." ICLR, 2024.

---

> ### Author Response · Authors · 2024-11-25
> **Response to Reviewer u9M1 (2/2)**
>
> Q2. Why are non-negative constraints generalized in different scenarios? It is unclear whether the same form of non-negative constraints is suitable for all tasks or if specific forms are needed for different scenarios.
>
> A2. Intuitively, it is widely believed that enhancing sparsity in neural networks can promote monosemanticity [1,2]. Meanwhile, to enhance sparsity, using the non-negative constraints, especially the ReLU activation function is a common technique [3]. Consequently, combining these motivations, we consider adding the non-negative constraints as a generalized method in different scenarios to enhance monosemanticity.
>
> To verify that non-negative constraints promote monosemanticity, we follow [3] and [4,5] to respectively evaluate a monosemantic score for vision and language models. **For vision models, we follow [3] and adopt semantic consistency as the metric.** The semantic consistency calculates the proportion of activated samples that belong to their most frequent class along each dimension. We evaluate the models in different cases, including the frozen models (Table 1, Figure 2) and the models fine-tuned with 20 % training samples and 20% noise rate (Figure 3).
>
> *Table1. Semantic consistency of different models.*
>
> |  | Poly (CL) | Mono (NCL) |
> | --- | --- | --- |
> | Linear Probing on CIFAR-100 | 1.0 | **8.2** |
> | Linear Probing on Imagenet-100 | 1.0 | **12.3** |
> |  | Poly (CE) | Mono (NCE) |
> | Fine-tuning with 20% Samples | 2.1 | **18.4** |
> | Fine-tuning with 20% Noisy Lables | 3.4 | **20.6** |
>
>  **As shown in the table above, Non-negative Contrastive Learning (NCL) and Non-negative tuning significantly improve the semantic consistency of trained models.** For language models, as we have no access to labels, current works [2,3] usually use sparsity as a surrogate metric to evaluate the monosmenaticity. Specifically, we evaluate the sparsity (zero value ratio) of the intermediate activations of the LoRA and MonoLoRA models, which is the intrinsic sparsity of the LoRA module.
>
> *Table 2. Evaluation of monosemanticity on LoRA and MonoLoRA with Llama-2-7B-Chat on SST2 and Dolly.*
>
> |  |  | Alignment Sparsity | Task Sparsity |
> | --- | --- | --- | --- |
> | SST2 | LoRA | 0 | 0 |
> |  | MonoLoRA | 45.54 | 36.71 |
> | Dolly | LoRA | 0 | 0 |
> |  | MonoLoRA | 38.69 | 40.00 |
>
>  As shown in the table above, MonoLoRA significantly improves the sparsity of trained models, which indicates that our method also enhances the monosemanticity of large language models.
>
> Theoretically, [3] has proved that the optimal solution of non-negative contrastive learning is monosemantic. We extend the theoretical framework in Appendix B.5 and prove that the solutions of non-negative tuning are also monosemantic, which further verifies the effectiveness of monosemanticity.
>
> **As for the implementations of non-negative constraints,** we select ReLU following [3]. And we conduct additional experiments with different activation functions Softplus and GeLU.
>
> *Table2. Linear probing accuracy (%) of CL and NCL implemented with different activation functions on ImageNet-100 under 0% and 90% label noise.*
>
> | Noise Rate | 0 | 90 |
> | --- | --- | --- |
> | CL | 66.8 | 34.4 |
> | NCL (ReLU) | 66.7 (-0.1) | 48.1 (+13.7) |
> | NCL (Softplus) | 57.4 (-9.4) | 47.1 (+12.7) |
> | NCL (GeLU) | 65.5 (-1.3) | 46.5 (+12.1) |
>
> As shown in the table above, taking linear probing under label noise as an example, different activation functions also show significant benefits in robustness performance. Meanwhile, the non-negative constraints with ReLU achieve the best empirical performance.
>
> Reference
>
> [1] Elhage, et al. "Softmax Linear Units." Transformer Circuits Thread, 2022.
>
> [2] Hoagy Cunningham, Aidan Ewart, Logan Riggs, Robert Huben, and Lee Sharkey. "Sparse autoen
> coders find highly interpretable features in language models." ICLR, 2024.
>
> [3] Wang, Yifei, et al. "Non-negative Contrastive Learning." ICLR, 2024.
>
> [4] Yan, Hanqi, et al. "Encourage or Inhibit Monosemanticity? Revisit Monosemanticity from a Feature Decorrelation Perspective." EMNLP 2024.
>
> [5] Gurnee, Wes, et al. "Finding neurons in a haystack: Case studies with sparse probing." arXiv preprint arXiv:2305.01610 (2023).
>
> ---
>
> Q3.  Abbreviations
>
> A3. Thanks for your suggestions and we have introduced the abbreviations at their first occurrence.
>
> ---
>
> Thank you again for your encouraging comments and valuable feedbacks. We have supplemented additional experiments and discussions following your suggestions, and we are keen to know whether you find it satisfactory. We are very happy to address your remaining concerns on our work.

---

> ### Author Response · Authors · 2024-11-27
>
> Dear Reviewer u9M1,
>
> We have carefully prepared a detailed response to address each of your questions. Would you please take a look and let us know whether you find it satisfactory?
>
> We note that Reviewer JMHa has appreciated our response and raised the score beyond the acceptance bar. We also respectfully suggest that you could re-evaluate our work with the updated explanations and results.
>
> Thanks! Have a great day!
>
> Authors

---

> ### Author Response · Authors · 2024-12-02
> **Your further inputs are greatly appreciated. Only 24 hours left.**
>
> Dear Reviewer u9M1,
>
> For your raised questions, we prepared a detailed response to address your concerns. We were hoping to hear your feedback on them.
>
> As there are only 24 hours left for the reviewer and author discussions and we understand that everyone has a tight schedule, we kindly wanted to send a gentle reminder to ensure that our response sufficiently addressed your concerns or if there are further aspects we need to clarify.
>
> If you could find the time to provide your thoughts on our response, we would greatly appreciate it.
>
> Best, Authors

---

> > ### Comment · Reviewer_u9M1 · 2024-12-03
> >
> > Thank you for your response. After reviewing the feedbacks, several of my concerns have been addressed. Consequently, I have decided to raise the score to 6.

---

### Official Review · Reviewer_bevX · 2024-11-04

**Soundness:** 3
**Presentation:** 2
**Contribution:** 3
**Rating:** 6
**Confidence:** 3

**Summary:**

This paper challenges the "accuracy-interpretability" tradeoff in deep learning models by considering the robustness of features across a range of approaches to learning. Experimental results validate that learned monosemantic features are more robust to noisy data and produces better performance when finetuning. The paper extends a toy model from Elhage et al. to suggest an explanation for this phenomenon.

**Strengths:**

1. The application of robustness as a performance metric to feature semanticity is a useful contribution.
2. The choice of baseline and methods to produce monosemanticity are well chosen.
3. The toy model provides a clear demonstration of why monosemantic features fare better.

**Weaknesses:**

1. The paper spends little time contextualising the "widely accepted belief" that there is an accuracy-interpretability tradeoff.
2. It isn't immediately clear why MonoLoRA promotes feature monosemanticity. A clearer explanation of the mathematics would be helpful.
3. The theoretical explanation is limited to label noise only, rather than finetuning scenarios. It would be good to address it at least in passing, or as a possible avenue of further work.

**Questions:**

1. What is the principled reason for distinguishing between few shot and LLM finetuning?
2. Should we expect a difference in outcomes between adding Gaussian noise and adding uniform noise?

---

> ### Author Response · Authors · 2024-11-25
> **Response to Reviewer bevX (1/2)**
>
> Thanks for your careful reading and critical review. Following your suggestions, we have added more discussions on the accuracy-interpretability tradeoff. We further address each of your concerns below, and hope you find them satisfactory.
>
> ---
>
> Q1. The paper spends little time contextualising the "widely accepted belief" that there is an accuracy-interpretability tradeoff.
>
> A1. Thanks for your suggestions and we have added an additional discussion on the accuracy-interpretability trade-off in Lines 43-47. To be specific, the modified paragraph is in the following:
>
> However, these works on monosemanticity suggest an inevitable ``accuracy-interpretability'' tradeoff: monosemantic features, although more interpretable, come at the sacrifice of expressive power and underperform polysemantic features at prediction accuracy. For example (Gao et al., 2024) observe that the sparse autoencoders significantly enhance the monosemanticity of language models while obtaining inferior performance in downstream prediction tasks.  (Elhage et al., 2022b)  show that monosemantic representations indicate lower model performance in reconstruction tasks with a toy model. ((Cunningham et al. 2023) exhibit a negative correlation between the expressive power and monosemanticity.
>
> ---
>
> Q2. Why MonoLoRA promotes feature monosemanticity.
>
> A2. In fact, it is widely discussed that enhancing activation sparsity in language models can promote monosemanticity [1, 2]. Besides, to enhance sparsity, using the non-negative constraints, especially the ReLU activation function is a common technique and has theoretical guarantees [3]. Consequently, combining these motivations, we consider adding the non-negative constraints and propose MonoLoRA. To verify that MonoLoRA promotes monosemanticity, we evaluate sparsity as a surrogate metric following [4,5,6]. Specifically, we evaluate the sparsity (zero value ratio) of the intermediate activations of the LoRA and MonoLoRA models, which is the intrinsic sparsity of the LoRA module.
>
> *Table 1. Evaluation of monosemanticity on LoRA and MonoLoRA with Llama-2-7B-Chat on SST2 and Dolly.*
>
> |  |  | Alignment Sparsity | Task Sparsity |
> | --- | --- | --- | --- |
> | SST2 | LoRA | 0 | 0 |
> |  | MonoLoRA | 45.54 | 36.71 |
> | Dolly | LoRA | 0 | 0 |
> |  | MonoLoRA | 38.69 | 40.00 |
>
> Results in the table above show that MonoLoRA significantly enhances the activation sparsity, which further verifies that MonoLoRA promotes feature monosemanticity.
>
> **Reference**
>
> [1] Elhage, et al. "Softmax Linear Units". Transformer Circuits Thread, 2022.
>
> [2] Hoagy Cunningham, Aidan Ewart, Logan Riggs, Robert Huben, and Lee Sharkey. "Sparse autoen
> coders find highly interpretable features in language models." ICLR, 2024.
>
> [3] Wang, Yifei, et al. "Non-negative Contrastive Learning." ICLR, 2024.
>
> [4] Elhage, Nelson, et al. "Toy models of superposition." arXiv preprint arXiv:2209.10652 (2022).
>
> [5] Yan, Hanqi, et al. "Encourage or Inhibit Monosemanticity? Revisit Monosemanticity from a Feature Decorrelation Perspective." EMNLP 2024.
>
> [6] Gurnee, Wes, et al. "Finding neurons in a haystack: Case studies with sparse probing." arXiv preprint arXiv:2305.01610 (2023).
>
> ---
>
> Q3.The theoretical explanation is limited to label noise only, rather than finetuning scenarios. It would be good to address it at least in passing, or as a possible avenue of further work.
>
> A3. In fact, it is difficult to analyze the relationship between the properties of representations and model performance during the finetuning scenarios. The existing theoretical framework of representation learning [1,2] mainly focuses on the performance of the frozen pretrained representations for the simplicity of analysis. So we first consider borrowing the idea from linear discriminant analysis and analyze the decision boundaries of polysemantic and monosemantic representations under noise (label noise (main paper) and random input noise (Appendix B.3)). **However, we totally agree that it is a valuable direction to analyze finetuning scenarios in further work and we highlight it in Line 501-505 in the revision.**
>
> Reference
>
> [1] Arora, Sanjeev, et al. "A theoretical analysis of contrastive unsupervised representation learning." arXiv preprint arXiv:1902.09229 (2019).
>
> [2 ]HaoChen, Jeff Z., et al. "Provable guarantees for self-supervised deep learning with spectral contrastive loss." NeurIPS 2021.

---

> ### Author Response · Authors · 2024-11-25
> **Response to Reviewer bevX (2/2)**
>
> Q4. What is the principled reason for distinguishing between few-shot and LLM finetuning?
>
> A4. In fact, since LLMs do not have a natural representation space like visual models, current research that discusses the monosemanticity of language models usually focuses on the neurons inside the networks [1,2].  In contrast, monosemanticity in vision models is often discussed and evaluated at the output representation layer [3]. Accordingly, when examining the relationship between monosemanticity and robustness in this paper, we apply non-negative constraints and sparse autoencoders to the representation layers of vision models while implementing MonoLoRA on the inside neurons.
>
> Reference
>
> [1] Gao, Leo, et al. "Scaling and evaluating sparse autoencoders." arXiv preprint arXiv:2406.04093 (2024).
>
> [2] Hoagy Cunningham, Aidan Ewart, Logan Riggs, Robert Huben, and Lee Sharkey. "Sparse autoen
> coders find highly interpretable features in language models." ICLR, 2024.
>
> [3] Wang, Yifei, et al. "Non-negative Contrastive Learning." ICLR, 2024.
>
> ---
>
> Q5. Should we expect a difference in outcomes between adding Gaussian noise and adding uniform noise?
>
> A5. In fact, although both of them are random noises added to the images, they influence the distribution of trained models in distinct ways. Consequently, as shown in Figure 2, the noise with similar strengths shows significantly different influences on models, e.g., the uniform noise tends to be more aggressive. So we evaluate both of them to verify that attaining monosemantcity enhances model robustness under different forms of input noise, and the consistent accuracy improvement brought by enhancing monosemanticity supports our claims.
>
> ---
>
> Thank you again for your encouraging comments and valuable feedback. We are very happy to address your remaining concerns on our work.

---

> ### Author Response · Authors · 2024-11-27
>
> Dear Reviewer bevX,
>
> We have carefully prepared a detailed response to address each of your questions. Would you please take a look and let us know whether you find it satisfactory?
>
> Thanks! Have a great day!
>
> Authors

---

> > ### Comment · Reviewer_bevX · 2024-11-28
> >
> > Thanks for the responses! It'll be great to have these in the next version of the paper.

---

> > > ### Author Response · Authors · 2024-12-02
> > >
> > > Dear Reviewer bevX,
> > >
> > > Thanks for your responses! We will definitely incorporate the additional results and discussions in the revision.
> > >
> > > Authors

---

### Official Review · Reviewer_1weT · 2024-11-06

**Soundness:** 3
**Presentation:** 4
**Contribution:** 4
**Rating:** 6
**Confidence:** 4

**Summary:**

The paper aims to investigate the role of monosemanticity of model features on model robustness. Monosemanticity of model features is characterised by individual dimensions of the model embedding layer being activated with single latent natural concepts (vs. several, which corresponds to polysemanticity of model features). Robustness is investigated empirically in linear probing experiments on CIFAR-100 and ImageNet-100 with respect to label noise, gaussian input noise, uniform input noise, and sketch and stylised distribution shifts. Furthermore, it is investigated empirically in few-shot finetuning on ImageNet-100 as well as on LoRA training for LLMs.
To analyse the effect of monosemanticity on robustness, models trained to exhibit monosemantic features (e.g. through sparse auto encoders) are compared to vanilla trained models (representing polysemantic feature models). In all comparisons where robustness is investigated as described, the models trained to exhibit monosemantic features perform better.
Lastly, the paper illustrates benefits of monosemanticity for model robustness on toy examples as well.

**Strengths:**

- The paper is investigating an interesting and relevant research question: is monosemanticity beneficial beyond interpretability, and for model robustness in particular?
- The paper is overall well written
- The paper has a good range of experiment beds (but individual experiments should be enriched, see below)

**Weaknesses:**

- Main weakness: At no point in Section 3 does the paper measure how ‘monosemantic’ the trained models are (according to their definition in line 114). So any claim of ‘monosemanticity -> robustness’ cannot be derived from the experiments in Section 3, as we do not know how monosemantic the models really are. Is there actually a quantifiable and significant difference in the monosemanticity of the models in Table 1, Figure 2, or Figure 3, or Table 2 (here, sparsity is given as a proxy for monosemanticity, but how close of a proxy it is is unclear)
- The definition of the polysemantic feature $v = x_1 - x_2$ in the theoretical toy example is specific rather than general, and insights derived from it will be accordingly so. For a more general validity of the results derived here, it should be of the form $w_1 x_1 + w_2 x_2 + b$ I believe.
- Figure 3a and 3b: it is possible that the cause for lower training and better validation accuracy is simply the regularisation and not the resulting monosemanticity. To understand this better, including a different regularisation that does not promote semanticity would be beneficial (e.g. early stopping or L2 regularisation, or even something like a negative-cross entropy). In any case monosemanticity of final models needs to be evaluated to draw conclusions here (see first weakness)
- Even if the other weaknesses were addressed, the paper would slightly overclaim, in particular in saying ‘feature monosematnicity to bring clear gains in model accuracy’ (l 100 and similar at other parts). As can be seen in the ‘0’ column of Table 1, monosemanticity is still at odds with model accuracy on clean iid accuracy (even though only slightly). Authors should specify accordingly that monosematnicity can bring gains in model robustness, not in general accuracy (as in some cases as the above it does not).
- For the vision model results, only a single backbone (ResNet18) is used in experiments. To give the claims and findings greater generality, including results of the same experiment suite but with a ViT backbone would be beneficial.

- Minor: typo l 242: Cifar-10 should be Cifar-100; typo l 257: ‘if there IS only a SMALL amount of …’,
- Minor: align references in l 120 and l 45

**Questions:**

-

---

> ### Author Response · Authors · 2024-11-25
> **Response to Reviewer 1weT (1/2)**
>
> Thanks for your careful reading and appreciating the presentation and contribution of our work.  We understand that your main concern lies in the lack of evaluating monosemanticity in our experiments. Below, we will elaborate on the supplemental evaluation and address your concerns point by point.
>
> ---
>
> Q1 The evaluation on monosemanticity of different models.
>
> A1. Thank you for pointing this out and we fully understand your concerns. We evaluate the monosemanticity of different models with current metrics. **For vision models, we follow [1] and adopt semantic consistency as the metric.** The semantic consistency calculates the proportion of activated samples that belong to their most frequent class along each dimension. We evaluate the models in different cases, including the frozen models (Table 1, Figure 2) and the models fine-tuned with 20 % training samples and 20% noise rate (Figure 3).
>
> *Table1. Semantic consistency of different models.*
>
> |  | Poly (CL) | Mono (NCL) | Mono (SAE) |
> | --- | --- | --- | --- |
> | Linear Probing on CIFAR-100 | 1.0 | **8.2** | 3.1 |
> | Linear Probing on Imagenet-100 | 1.0 | **12.3** | 7.2 |
> |  | Poly (CE) | Mono (NCE) |  |
> | Fine-tuning with 20% Samples | 2.1 | **18.4** |  |
> | Fine-tuning with 20% Noisy Lables | 3.4 | **20.6** |  |
>
>  **As shown in the table above, Non-negative Contrastive Learning (NCL), Sparse Autoencoder (SAE), and Non-negative tuning significantly improve the semantic consistency of trained models,** which further supports our claim that attaining monosemanticity can enhance model robustness. Besides, we also note that the NCL obtains larger improvements on monosemanticity. which is consistent with the results that NCL performs better than SAE under noise.
>
> For language models, as we have no access to labels, current works [2,3] usually use sparsity as a surrogate metric to evaluate the monosmenaticity. Specifically, we evaluate the sparsity (zero value ratio) of the intermediate activations of the LoRA and MonoLoRA models, which is the intrinsic sparsity of the LoRA module.
>
> *Table 2. Evaluation of monosemanticity on LoRA and MonoLoRA with Llama-2-7B-Chat on SST2 and Dolly.*
>
> |  |  | Alignment Sparsity | Task Sparsity |
> | --- | --- | --- | --- |
> | SST2 | LoRA | 0 | 0 |
> |  | MonoLoRA | 45.54 | 36.71 |
> | Dolly | LoRA | 0 | 0 |
> |  | MonoLoRA | 38.69 | 40.00 |
>
>  As shown in the table above, MonoLoRA significantly improves the sparsity of trained models, which indicates that our method also enhances the monosemanticity of large language models.
>
> **Reference**
>
> [1] Wang, Yifei, et al. "Non-negative Contrastive Learning." ICLR, 2024.
>
> [2] Yan, Hanqi, et al. "Encourage or Inhibit Monosemanticity? Revisit Monosemanticity from a Feature Decorrelation Perspective." EMNLP 2024.
>
> [3] Gurnee, Wes, et al. "Finding neurons in a haystack: Case studies with sparse probing." arXiv preprint arXiv:2305.01610 (2023).
>
> ---
>
> Q2. The generalization of toy theoretical examples.
>
> A2. Thanks for your suggestion, we have revised our theoretical toy example and adopted a more general form (replacing $\nu = x_1 - x_2$ with $\nu = w_1x_1 - w_2x_2$). As shown in Theorem B.13, the results remain consistent, which still implies that the monosemantic representations indicate more robust performance under label noise and input noise.
>
> ---
>
> Q3. Is it possible that the cause for lower training and better validation accuracy is simply the regularisation and not the resulting monosemanticity?
>
> A3. Indeed, it is reasonable to suppose that the lower training and better validation accuracy result from the regularization. To verify whether the benefits come from enhancing monosemanticity, we consider the early-stoping techniques following your suggestions. Specifically, we fine-tune the models with 20% training samples for 100 epochs as an example. We respectively evaluate the accuracy on 60, 80, and 100 epochs.
>
> *Table 3. Fine-tuning with early stopping on ImageNet-100 using 20% training samples.*
>
> | Stop Epoch | Semantic Consitency |  | Trainining Accuracy |  | Validation Accuracy |  |
> | --- | --- | --- | --- | --- | --- | --- |
> |  | Poly (CE) |  Mono (NCE) | Poly (CE)  | Mono (NCE)  | Poly (CE) | Mono (NCE) |
> | 60 | 1.5 | **16.7** | **78.1** | 73.7 | 55.2 | **58.1** |
> | 80 | 1.9 | **17.5** | **88.9** | 85.1 | 60.4 | **62.0** |
> | 100 | 2.1 | **18.4** | **89.6** | 86.1 | 60.6 | **62.7** |
>
> As shown in the table, the simple regularization technique that does not promote semantic consistency (i.e., monosemanticity) fails to improve validation performance. This further confirms that the benefits come from enhancing monosemanticity rather than simple regularization.

---

> ### Author Response · Authors · 2024-11-25
> **Response to Reviewer 1weT(2/2)**
>
> Q4. Saying ‘feature monosematnicity to bring clear gains in model accuracy’ overclaims.
>
> A4. Thanks for your suggestions and we have revised our claims in our revision, including Line 18, Line 100, and Line 425. To be specific, we highlight that attaining monosemanticity brings significant improvements in tasks related to model robustness.
>
> ---
>
> Q5. Including results of the same experiment suite but with a ViT backbone would be beneficial.
>
> A5. Thanks for your suggestions, we supplement additional experiments on ViT backbones. To be specific, we consider training ViT-Small respectively with contrastive learning and non-negative contrastive learning as an example. We then evaluate the linear probing performance under different noise rates.
>
> *Table 4. Linear probing accuracy and gains (%) of ViT-small trained with CL and NCL on ImageNet-100 under different noise rates (%).*
>
> |  | 0 | 30 | 60 | 90 |
> | --- | --- | --- | --- | --- |
> | CL | **58.2** | 48.6 | 37.8 | 10.6 |
> | NCL | 58.1 | **50.8** | **43.2** | **13.5** |
> | Gains | -0.1 | +2.2 | +5.4 | +2.9 |
>
> As shown in the tables above, enhancing monosemanticity in ViT backbones improves the performance under noise, which further verifies our claims that attaining monosemanticity enhances model robustness.
>
> ---
>
> Q6. typos in Lines 45,120, 242, and 257.
>
> A6. Thanks for your suggestions and we have fixed the typos in the revision.
>
> ---
>
> Thank you again for your valuable suggestions. We have conducted additional evaluations of our experiments, and address each of your concerns above. We respectfully suggest that you could re-evaluate our work based on these updated results. We are very happy to address your remaining concerns about our work.

---

> ### Author Response · Authors · 2024-11-27
>
> Dear Reviewer 1weT,
>
> We have carefully prepared a detailed response to address each of your questions. Would you please take a look and let us know whether you find it satisfactory?
>
> We note that Reviewer JMHa has appreciated our response and raised the score beyond the acceptance bar. We also respectfully suggest that you could re-evaluate our work with the updated explanations and results.
>
> Thanks! Have a great day!
>
> Authors

---

> ### Comment · Reviewer_1weT · 2024-11-27
>
> With the additional results provided, the paper seems much stronger to me, and I have updated my score accordingly. I believe the additional results in Appendix C.2 / Table 3 should be in the main part of the paper as they are core to the paper's argument.

---

> > ### Author Response · Authors · 2024-12-02
> >
> > Dear Reviewer 1weT,
> >
> > Thanks for your responses! We are glad to hear that our previous replies have addressed your concerns. We will definitely incorporate the additional results in Table 3 as the main part of the revision.
> >
> > Authors

---

### Author Response · Authors · 2024-11-25
**Paper Update**

We sincerely thank all reviewers for their detailed reviews and valuable comments. We have carefully responded to their concerns and incorporated these suggestions in the updated manuscript. The main revisions include:

1. Additional discussions on the interpretability-accuracy trade-off in Section 1.
2. Additional discussions to clear our contributions and novelty in Section 2.
3. The report of mean and standard deviation results obtained from random runs for empirical results in Section 3.
4. Additional theoretical analysis in the toy models with a more generalized form, i.e., we replace $\nu = x_1 - x_2$ with $\nu = w_1x_1 - w_2x_2$ in Appendix B.4.
5. Additional theoretical analysis to verify that the representations learned with non-negative constraints are monosemantic in Appendix B.5.
6. **Additional evaluation on monosematicity of different models (used in Table 1, Figure 2, Figure 3, Table 2) in Appendix C.2.**

Below, we will address the concerns of the reviewers point by point in the response to each reviewer.

---

### Comment · Area_Chair_a8BB · 2024-11-25

Dear Reviewers,

This is a friendly reminder that the authors have submitted a rebuttal and the end of the discussion period is November 26th AoE. Due to the tight deadline please take a look at it as soon as possible and check if your questions/comments have been properly addressed. At the very least, please acknowledge that you have read the authors' response to your review.

Thank you for your time and effort!

AC

---

### Comment · Area_Chair_a8BB · 2024-11-27

Dear Reviewers,

We wanted to let you know that the discussion period has been extended to December 2nd. If you haven't had the opportunity yet, we kindly encourage you to read the rebuttal at your earliest convenience and verify whether your questions and comments have been fully addressed.

We sincerely appreciate your time, effort, and thoughtful contributions to this process.

Best,

AC

---

### Meta-Review · Area_Chair_a8BB · 2024-12-14

**Metareview:**

This paper challenges the idea that making AI models more interpretable makes them less accurate. The authors show that forcing neurons in deep learning models to become monosemantic not only improves interpretability but also makes the models more robust across various challenges like noisy data, limited training examples, and such.

This suggests an interesting link between interpretability and robustness.

After reading the reviews I see a common concern on the clarity of this work. However, I believe this concern has been addressed during the rebutal, and this has been reflected in the reviewer's scores. There was also the concern of measuring monosemanticity in the models (1weT, bevX, JMHam), for wich the authors have introduced an evaluation using semantic consistency as the metric.

Overall, this work is leaning towards borderline accept and I believe it brings interesting insights to the research community, thus I recommend its acceptance.

**Additional Comments On Reviewer Discussion:**

After the rebuttal, reviewers bevX, 19M1, JMha and 1weT, recommend ""6: marginally above borderline"", while qpFh recommends this work with: ""5: marginally below borderline"". However, reviewer qpFh has not yet acknowledged the author's rebuttal, which I believe succeeds at resolving at least some of the reviewer's concerns.

---

### Decision · Program_Chairs · 2025-01-22

Accept (Poster)